# HNRNPM controls circRNA biogenesis and splicing fidelity to sustain cancer cell fitness

Jessica SY Ho[1,2], Federico Di Tullio[3,4,5], Megan Schwarz[3,4,5], Diana Low[1], Danny Incarnato[6,7,8], Florence Gay[1], Tommaso Tabaglio[1], JingXian Zhang[1], Heike Wollmann[1], Leilei Chen[9,10], Omer An[9], Tim Hon Man Chan[9], Alexander Hall Hickman[1], Simin Zheng[2,11], Vladimir Roudko[4,5], Sujun Chen[12,13,14], Alcida Karz[15,16], Musaddeque Ahmed[13], Housheng Hansen He[12,13], Benjamin D Greenbaum[4,5,17,18], Salvatore Oliviero[6,7], Michela Serresi[19], Gaetano Gargiulo[19], Karen M Mann[20], Eva Hernando[15,16], David Mulholland[4,5], Ivan Marazzi[2], Dave Keng Boon Wee[1], Ernesto Guccione[1,3,4,5]*

[1]Institute of Molecular and Cell Biology (IMCB), Agency for Science, Technology and Research (A*STAR), Singapore, Singapore; [2]Department of Microbiology, Icahn School of Medicine at Mount Sinai, New York, United States; [3]Center for Therapeutics Discovery, department of Oncological Sciences and Pharmacological Sciences, Tisch Cancer Institute, Icahn School of Medicine at Mount Sinai, New York, United States; [4]Tisch Cancer Institute, Icahn School of Medicine at Mount Sinai, New York, United States; [5]Department of Oncological Sciences, Icahn School of Medicine at Mount Sinai, New York, United States; [6]IIGM (Italian Institute for Genomic Medicine), Torino, Italy; [7]Dipartimento di Scienze della Vita e Biologia dei Sistemi Università di Torino, Torino, Italy; [8]Department of Molecular Genetics, Groningen Biomolecular Sciences and Biotechnology Institute (GBB), University of Groningen, Groningen, Netherlands; [9]Cancer Science Institute of Singapore, National University of Singapore, Singapore, Singapore; [10]Department of Anatomy, Yong Loo Lin School of Medicine, National University of Singapore, Singapore, Singapore; [11]NTU Institute of Structural Biology, Nanyang Technological University, Singapore, Singapore; [12]Department of Medical Biophysics, University of Toronto, Toronto, Canada; [13]Princess Margaret Cancer Center, University Health Network, Toronto, Canada; [14]Ontario Institute for Cancer Research, Toronto, Canada; [15]Interdisciplinary Melanoma Cooperative Group, New York University Langone Medical Center, New York, United States; [16]Department of Pathology, New York University Langone Medical Center, New York, United States; [17]Department of Medicine, Hematology and Medical Oncology, Icahn School of Medicine at Mount Sinai, New York, United States; [18]Department of Pathology, Icahn School of Medicine at Mount Sinai, New York, United States; [19]Max Delbruck Center for Molecular Medicine, Berlin-Buch, Germany; [20]Department of Molecular Oncology, Moffitt Cancer Center, Tampa, United States

*For correspondence:
ernesto.guccione@mssm.edu

Competing interests: The authors declare that no competing interests exist.

**Abstract** High spliceosome activity is a dependency for cancer cells, making them more vulnerable to perturbation of the splicing machinery compared to normal cells. To identify splicing factors important for prostate cancer (PCa) fitness, we performed pooled shRNA screens in vitro and in vivo. Our screens identified heterogeneous nuclear ribonucleoprotein M (HNRNPM) as a

regulator of PCa cell growth. RNA- and eCLIP-sequencing identified HNRNPM binding to transcripts of key homeostatic genes. HNRNPM binding to its targets prevents aberrant exon inclusion and backsplicing events. In both linear and circular mis-spliced transcripts, HNRNPM preferentially binds to GU-rich elements in long flanking proximal introns. Mimicry of HNRNPM-dependent linear-splicing events using splice-switching-antisense-oligonucleotides was sufficient to inhibit PCa cell growth. This suggests that PCa dependence on HNRNPM is likely a result of mis-splicing of key homeostatic coding and non-coding genes. Our results have further been confirmed in other solid tumors. Taken together, our data reveal a role for HNRNPM in supporting cancer cell fitness. Inhibition of HNRNPM activity is therefore a potential therapeutic strategy in suppressing growth of PCa and other solid tumors.

## Introduction

In eukaryotic cells, the majority of genes harbor intronic sequences that are removed during RNA splicing and transcript maturation (*Venter et al., 2001*). This process is regulated by the spliceosome, comprising small non-coding RNAs (U1, U2, U4, U5, and U6), the core spliceosomal proteins (i.e., U2AF1, U2AF2, SF3B1, etc.), and other auxiliary factors (i.e., HNRNPs, SRSFs, etc.) (*Wahl et al., 2009*). Under normal physiological conditions, proper regulation of splicing provides the cell an opportunity to control gene expression in the absence of genetic alterations. By expressing alternative isoforms of the same gene, the cell can regulate both the inclusion/exclusion of specific protein and/or RNA domains (*Yang et al., 2016*; *Ellis et al., 2012*) or change transcript half-life (*Naro et al., 2017*; *'t Hoen et al., 2011*; *Lareau and Brenner, 2015*), thus impacting both transcript fate and function.

The regulation of alternative splicing plays a central role in development (*Baralle and Giudice, 2017*), cellular differentiation (*Pimentel et al., 2016*), as well as in the cellular response to external or internal stimuli (*Shalgi et al., 2014*; *Hang et al., 2009*; *Makino et al., 2002*; *Haque et al., 2018*). However, the same phenotypic plasticity offered by the splicing machinery can work against the cell and organism, and confer competitive advantages to cells under pathological conditions such as cancer. Indeed, alterations in the expression of specific isoforms of certain genes and of splicing factors themselves can promote cell proliferation (e.g., androgen receptor [AR]; *Liu et al., 2014*), metastasis (e.g., CD44; *Todaro et al., 2014*; *Xu et al., 2014*), or avoid apoptosis (*Dewaele et al., 2016*; *Schwerk and Schulze-Osthoff, 2005*) in cancer. Some of these changes in isoform ratios may be direct results of mutations in the linear motifs that comprise the splice sites (*Puente et al., 2015*; *Bartram et al., 2017*; *Jung et al., 2015*) or in the splicing machinery itself (*Yoshida et al., 2011*; *Chronic Myeloid Disorders Working Group of the International Cancer Genome Consortium et al., 2011*; *Wang et al., 2011*; *Graubert et al., 2011*).

In addition to the differential splicing of specific genes, there is also increasing evidence that some molecular subtypes of cancer, which bear mutations in non-splicing-related genes such as MYC, are highly dependent on a functional core spliceosome for survival (*Koh et al., 2015*; *Hsu et al., 2015*). This may be related to their high proliferation rates, which require them to heavily rely on the splicing machinery.

Given the strong involvement of RNA splicing in cancer, the splicing machinery and its associated factors may offer selective therapeutic vulnerabilities. However, the identification of critical targets, the context in which they function, their mechanisms of action, and the functional impact of the individual splicing factors in cancer remain largely unexplored.

To address this, we performed parallel pooled in vivo and in vitro shRNA screens against core and auxiliary splicing factors in prostate cancer (PCa). Through this screen, we were able to uncover a novel role for the heterogeneous nuclear ribonucleoprotein M (HNRNPM). HNRNPM expression is higher in PCa than in normal prostate epithelial cells (PrEC), and its loss affects PCa cell growth in vitro and in vivo. Our results have further been confirmed in other solid tumors (i.e., melanoma, lung adenocarcinoma, and pancreatic adenocarcinoma). RNA-seq and eCLIP suggest that HNRNPM binding differentially impacts the splicing and/or expression of distinct classes of transcripts. Interestingly, we found that HNRNPM limits circular RNA (circRNA) biogenesis within cells. Overall, our work offers mechanistic insight into the function of HNRNPM in prostate and other cancers, suggesting that HNRNPM could serve as a potential marker or drug target in cancer.

## Results

### A targeted pooled shRNA screen identifies splicing-related dependencies in PCa

In order to identify major splicing factor dependencies in PCa, we conducted parallel pooled in vitro and in vivo shRNA screens in the LNCAP PCa cell line (*Figure 1A*). In such a screen, shRNAs targeting essential genes and oncogenes are generally depleted as the cell population increases, whereas shRNAs that target tumor suppressors and negative regulators of proliferation become enriched. The screens utilized a lentiviral library of 520 shRNAs targeting 102 core and auxiliary splicing factors, meaning 3–5 shRNAs targeting each gene (*Supplementary file 1*). After puromycin selection, we harvested a portion of the cells to be used as the control input pool (P1). The remaining cells were either passaged continuously in vitro for about 28–30 population doublings (about 1 month; approximately nine passages), or xenografted into the flanks of SCID mice and allowed to form tumors over time. In vitro passaged cells were harvested at every other passage for DNA isolation (P3, P5, P7, P9). In vivo tumors were harvested for DNA isolation when they attained a size of 400 mm$^3$ or more. shRNA hairpin enrichment levels in individual samples were quantified via high-throughput sequencing. Input lentiviral and plasmid pools were also sequenced as controls.

For analysis, the abundance of individual hairpins was normalized to the control input pool (P1). Multidimensional scaling (MDS) analysis of the data showed P1 clustering with the plasmid and lentiviral controls, confirming that hairpin abundances in our input pool of cells were representative of our initial library (*Figure 1B*). Tumor samples were clustered away from P1 samples, indicating a shift in overall hairpin distribution in these samples. Notably, for the in vitro passaged samples, those derived from the later passages clustered closer to tumor samples, while samples derived from the earlier passages clustered more closely to the input control (*Figure 1B*), possibly underscoring similarities in the functional enrichment/depletion patterns during cell proliferation. Consistently, we identified five well-defined clusters of hairpin enrichment/depletion, with similar patterns of enrichment/depletion in both in vitro passaged and in vivo tumor samples (*Figure 1—figure supplement 1A*).

To identify top hits in our screen, we subjected our dataset to a rotational gene set analysis (*Wu et al., 2010*; *Figure 1C*, *Figure 1—source data 1* and *Figure 1—source data 2*). This analysis considers the overall enrichment levels of individual hairpins targeting the same gene and compares them against each other. Using this analysis, we were able to identify a common set of seven genes (*HNRNPL, SYNCRIP, HNRNPM, HNRNPF, SF3A1, SRSF2,* and *SRSF1*) that were significantly enriched or depleted in both the in vitro and in vivo screens (*Figure 1C*). These hits included previously characterized oncogenic splicing factors *SRSF1* (*Karni et al., 2007*) and *HNRNPL* (*Fei et al., 2017*), further confirming the validity of our screen. The majority of hairpins in our screen appeared to be depleted over time both in vitro and in vivo (*Figure 1—figure supplement 1A*), suggesting that some of the hits we found could have essential functions within cells. To distinguish between essential genes from genes that had oncogenic function, we further performed a short-term (96 hr), independent siRNA screen (*Figure 1D*) for our top hits (two-sided p≤0.01, false discovery rate [FDR] ≤ 0.05) on normal, untransformed PrEC. We rationalized that an acute reduction in expression of essential genes would likely be deleterious in these cells. On the other hand, reduced expression of genes that were more likely to be important for oncogenic growth should minimally inhibit PrEC growth. Indeed, short-term, reduced expression of several genes, such as *SF3A1*, *SRSF1*, and *SRSF2*, which are essential genes that form part of the core spliceosomal machinery, was rapidly deleterious in the PrECs (*Figure 1D*). On the other hand, reduced expression of *HNRNPM* and *HNRNPF* mildly affected proliferation in the PrEC line, suggesting a long-term oncogenic role rather than an essential role in PCa (*Figure 1D*). To further narrow down the hits from our screen, we reanalyzed publicly available gene expression datasets for PCa (PRAD) generated by The Cancer Genome Atlas (TCGA). This analysis correlated with the increased expression of *HNRNPM* with poorer disease-free survival over time, while this was not the case for *HNRNPF* (*Figure 1—figure supplement 1B*). Analysis of *HNRNPM* mRNA expression in normal and tumor prostate adenocarcinoma (PRAD) patient samples in the same datasets suggested that HNRNPM mRNA expression levels were significantly higher in tumor samples when compared with normal controls (*Figure 1—figure supplement 1C*). Western blot analysis of PrEC and PCa cell lines, LNCAP and PC3 further corroborated these

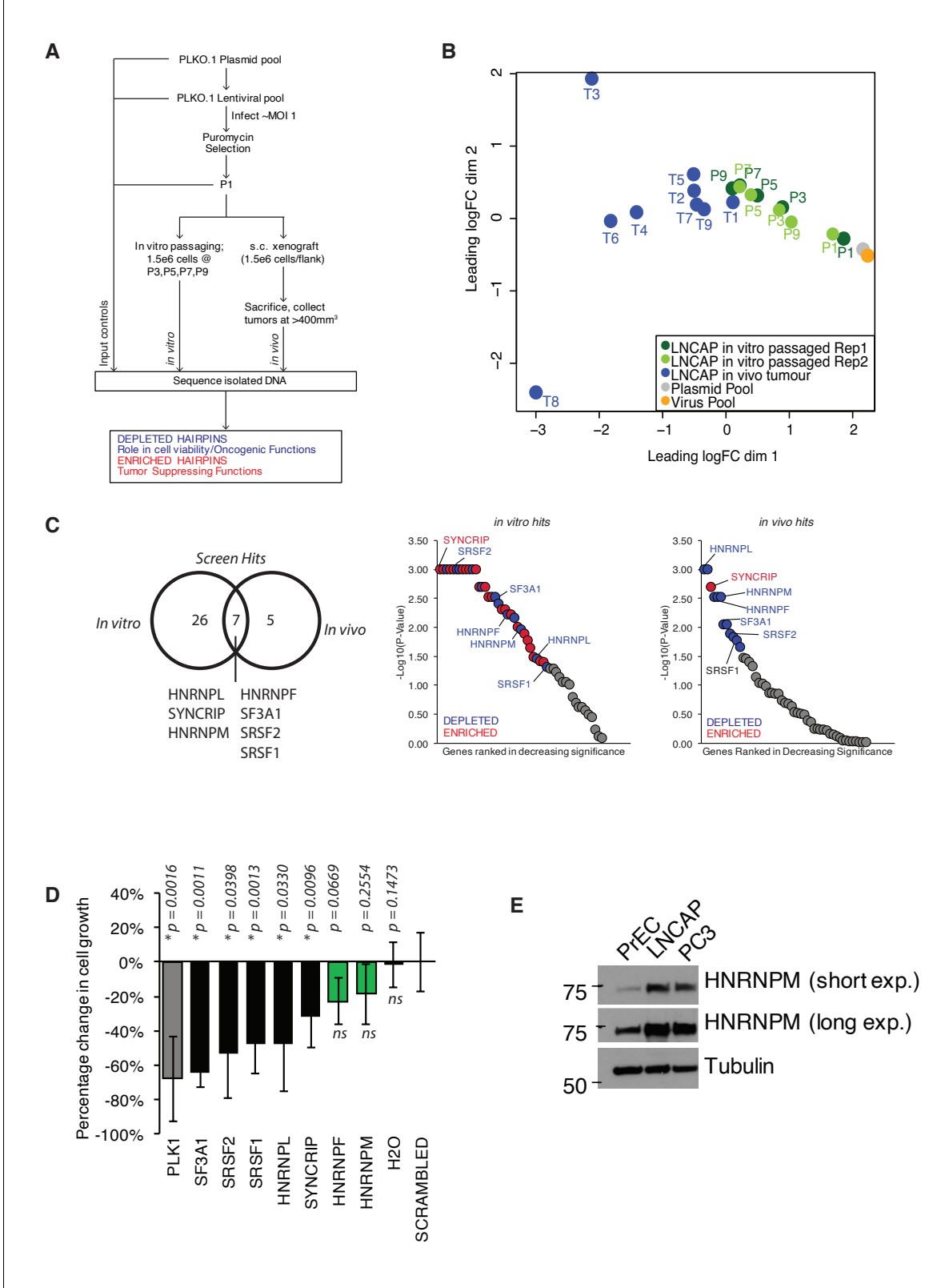

**Figure 1.** A pooled shRNA screen identifies HNRNPM as a regulator of prostate cancer (PCa) cell growth. (**A**) Schematic of overall experimental design. (**B**) Multidimensional scaling (MDS) plot of individual samples (in vitro passaged cells and tumors) collected in the screen, in relation to each other. (**C**) Right: Venn diagram showing overlap of significant (p<0.05) hits for in vitro and in vivo screens. Left: ranked dotplots showing –log10(p-values) of in vivo and in vitro screen hits. Red circles represent significantly upregulated (enriched) hits, while blue circles represent significantly downregulated

*Figure 1 continued on next page*

*Figure 1 continued*

(depleted) hits. Tumor-specific hits are indicated. (**D**) Barplots showing overall prostate epithelial cell (PrEC) proliferation upon a 96 hr, siRNA-mediated knockdown of the indicated splicing factors, as measured through a colorimetric MTS assay. siRNAs that inhibit cell proliferation relative to the scrambled siRNA control are indicated in black, while siRNAs that do not alter cell proliferation are indicated in green. Each assay was performed in two biological replicates, with two technical duplicates per replicate. Error bars represent SD; *p<0.05 compared to scrambled. (**E**) Western blot of HNRNPM protein levels in PrEC, LNCAP, and PC3 cells. Tubulin is shown as a loading control.

The online version of this article includes the following source data and figure supplement(s) for figure 1:

**Source data 1.** Rotational Gene Set Analysis (ROAST) results: tumor versus input (P1).
**Source data 2.** Rotational Gene Set Analysis (ROAST) results: in vitro.
**Figure supplement 1.** Verification of top hits in the pooled shRNA screen.

observations, showing that protein expression of *HNRNPM* was increased in PCa cells, but not in normal PrECs (*Figure 1E*). Taken together, *HNRNPM* is a valid target for further analysis.

## HNRNPM is required for PCa cell proliferation in vitro and in vivo

We first validated our screen results using shRNAs (TRCN0000001244; 2B7, TRCN0000001246; 2B9) present in the screen (*Figure 2A, B*, *Figure 2—figure supplement 1A*) in the LNCAP cell line. These hairpins were efficient in reducing HNRNPM protein and RNA levels (*Figure 2A , B*). A third hairpin we tested (TRCN0000001247; 2B10) did not reduce HNRNPM protein levels significantly and was discarded (*Figure 2—figure supplement 1B, C*). In line with the screen results, reduced *HNRNPM* mRNA and protein levels in LNCAP cells resulted in reduced cell proliferation (*Figure 2C*). Moreover, we showed that HNRNPM depletion impairs colony formation (*Figure 2D*) and anchorage-independent cell growth (*Figure 2E*), underscoring a broad role for *HNRNPM* in preserving cell fitness. *HNRNPM* dependency appeared similar in the PC3 PCa cell line (*Figure 2—figure supplement 1B–F*), which bears different driver mutations (*PTEN* null, *TP53* null, androgen independent) from LNCAP (*PTEN* null, *TP53* wildtype, androgen dependent) (*van Bokhoven et al., 2003*). We were also able to validate our findings in two independently derived human primary PCa cell lines (*Figure 2F*). This suggests that the impact of HNRNPM in PCa is not dependent on specific tumor genotypes or hormone dependency. Finally, the ability of LNCAP cells to form tumors when xenografted in mice was impaired in HNRNPM-depleted cells as compared to control cells (*Figure 2G*, *Figure 2—figure supplement 2*). Notably, tumor growth in these mice correlated to the potency of the shRNA used. Tumors treated with the less potent 2B7 shRNA (n = 11) were initially significantly smaller than scrambled shRNA treated tumors (n = 11); (*Figure 2G, Figure 2—figure supplement 2*, day 19), but escaped shRNA suppression by day 60 post injection (*Figure 2G*, *Figure 2—figure supplement 2*). On the other hand, tumors treated with the more potent 2B9 shRNA (n = 11) remained significantly smaller than scrambled treated tumors throughout the course of the experiment. The reduction in cell proliferation and growth correlated with potency of HNRNPM protein depletion by individual hairpins supports an on-target effect (*Figure 2*, *Figure 2—figure supplement 2*). These results suggest that HNRNPM is important for maintaining PCa cell growth both in vitro and in vivo.

## HNRNPM is bound to GU-rich elements within long genes

To further understand the molecular role of HNRNPM in PCa, we next performed eCLIP (*Van Nostrand et al., 2016*) on LNCAP cells to identify direct RNA targets of HNRNPM. eCLIP analysis (*Figure 3—figure supplement 1*; eCLIP controls) revealed 57,984 high-confidence (p-value≤0.05, fold enrichment ≥ 1.5) HNRNPM-bound sites in LNCAP cells (*Figure 3A*, *Figure 3—figure supplement 1*), spanning 6510 gene bodies. These sites were primarily located in intronic regions of expressed genes (92.3%), with very few binding sites found at either intergenic or exonic loci (7.8%; *Figure 3B*). Genes bound by HNRNPM were surprisingly much longer on average (p<2.2e-16; *Figure 3C*), with median lengths of 4181 bp, compared to the median lengths of 2988 bp for all expressed transcripts and 2447 bp for non-HNRNPM-bound and -expressed transcripts. HNRNPM-bound sites tended to be guanidine and uridine (GU)-rich (*Figure 3D, E*), and motif analyses of these sites revealed sequences consisting of single or double G residues interspersed with one to two U residues (*Figure 3E*).

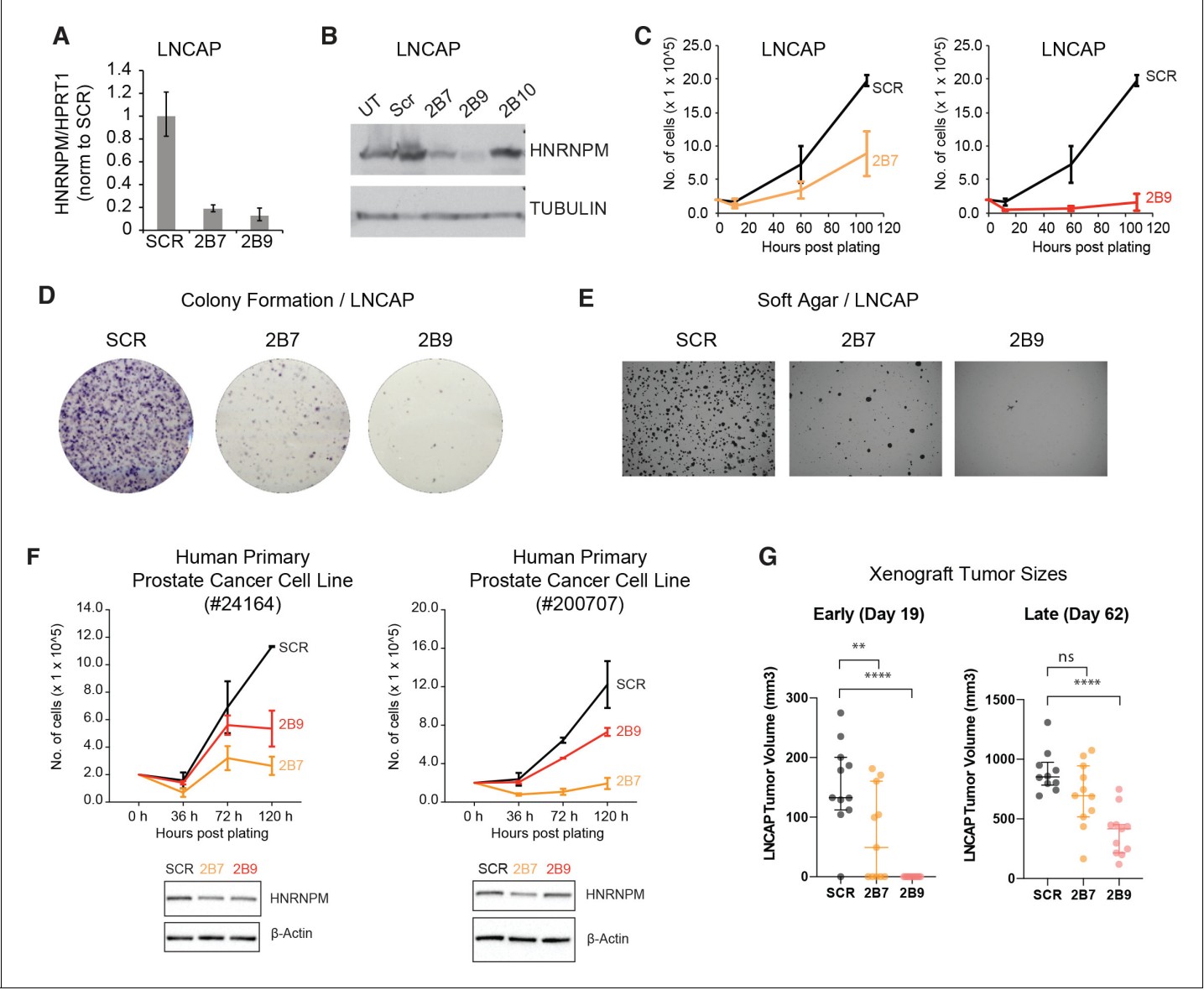

**Figure 2.** HNRNPM inhibits prostate cancer (PCa) cell growth in vitro and in vivo. HNRNPM RNA (**A**) and protein levels (**B**) upon expression of scrambled (Scr) or HNRNPM-specific shRNAs (2B7, 2B9, 2B10) in LNCAP cells. UT: untreated cells. (**C**) Cell proliferation assays of LNCAP cells expressing either scrambled or HNRNPM-specific shRNAs (2B7 and 2B9). Shown are the mean and standard deviation of two biological replicates. (**D**) Colony formation assays and (**E**) anchorage-independent growth (soft agar assays) of LNCAP cells expressing either scrambled or HNRNPM-specific shRNAs (2B7 and 2B9). Shown are representative scans of three independent replicates. (**F**) Cell proliferation (top panels) and western blot (lower panels) of two independently derived primary human PCa cell lines (#24164 and #200707) expressing either SCR or HNRNPM-specific shRNAs (2B7 and 2B9). Shown are the mean and standard deviation of two biological replicates. Protein expression of HNRNPM and β-actin at assay start is shown below. (**G**) Size of xenografted, SCR, or HNRNPM-specific shRNA (2B7, 2B9) expressing LNCAP tumors at 19 and 60 days post injection in 6–8-week-old CB17-SCID mice (n = 11 per condition). A one-way ANOVA/Dunnet's post-hoc test was used to calculate p-values. Ns: not significant. **p<0.01; ****p<0.0001.

The online version of this article includes the following figure supplement(s) for figure 2:

**Figure supplement 1.** Impact of HNRNPM deficiency in independent prostate cancer (PCa) cell lines.

**Figure supplement 2.** Xenograft tumor growth over time.

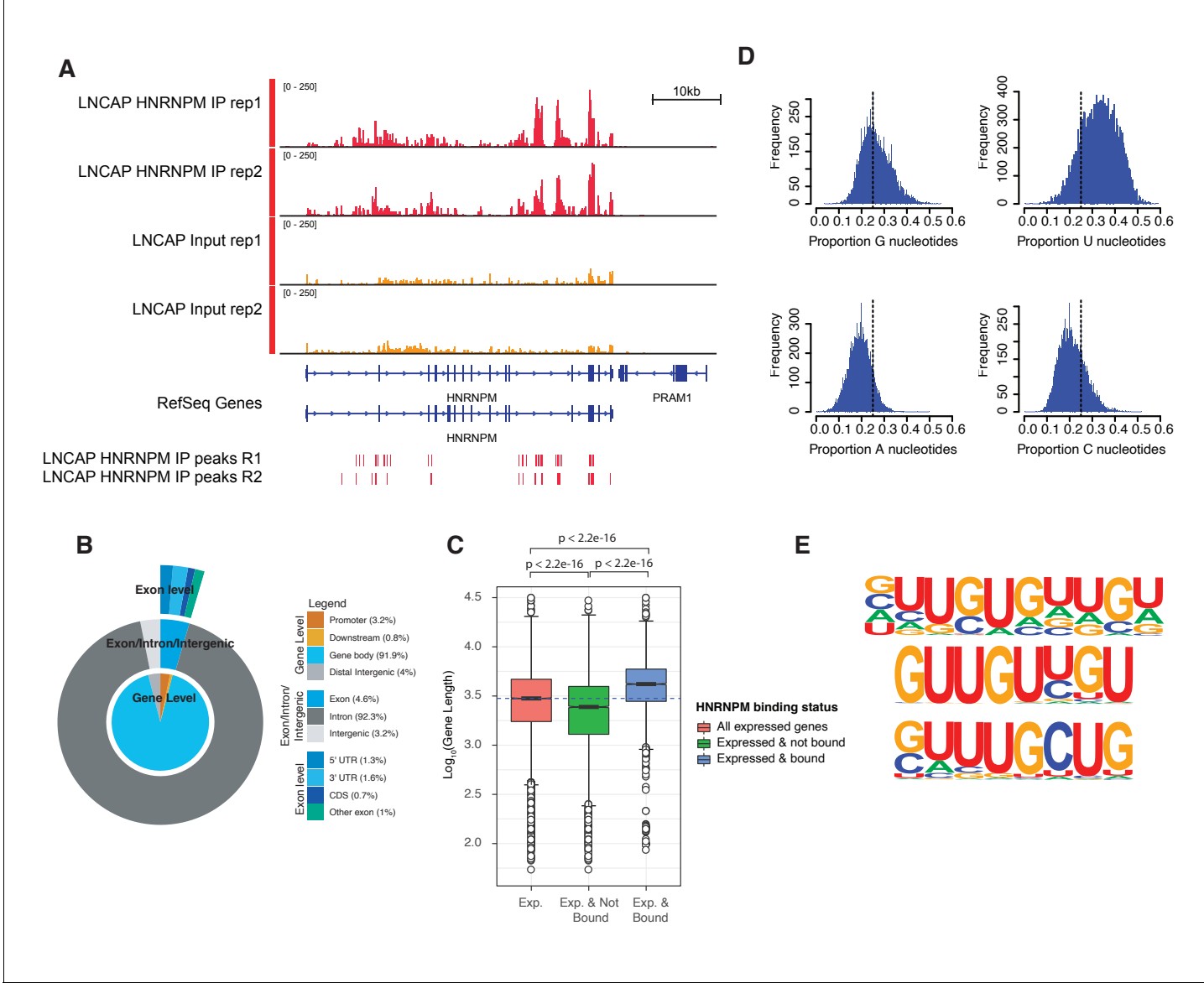

**Figure 3.** HNRNPM binds guanidine and uridine (GU)-rich elements within long introns. (**A**) Normalized read density of HNRNPM eCLIP at the HNRNPM and PRAM1 genes. Input (yellow) and immunoprecipitation (red) tracks are shown. High-confidence (p<0.05, fold enrichment >2) peaks are shown in the lowest track. (**B**) Distribution of HNRNPM-bound peaks across different genomic features. Relative frequency of total of specific events is indicated in the legend. (**C**) Distribution of gene lengths in all expressed genes (red), expressed and non-HNRNPM-bound genes (green) and expressed and HNRNPM-bound genes (blue). Log10(Gene length in base pairs) is plotted. (**D**) Histogram plot of nucleotide frequencies within called, high-confidence HNRNPM-bound peaks. (**E**) Motif analysis of HNRNPM-bound peaks. Shown are the top three (from top to bottom; p=1e-726, p=1e-625, and p=1e-597) motifs found within intragenic HNRNPM-bound sites.

The online version of this article includes the following figure supplement(s) for figure 3:

**Figure supplement 1.** eCLIP analysis of HNRNPM-binding sites (associated with *Figure 3*).

## Loss of HNRNPM results in minor changes in total mRNA levels

HNRNPM binding to pre-mRNA could potentially alter either transcript abundance or splicing. To better understand the impact of reduced HNRNPM binding in PCa cells, we further performed RNA sequencing on LNCAP PCa cells that were transduced either with the scrambled shRNA or HNRNPM-specific shRNAs (2B7 and 2B9; n = 3 replicates per shRNA) (*Figure 4A*, *Figure 4—source data 1*). When compared to scrambled shRNA control-treated cells, the overall transcript abundance shifts in both populations of HNRNPM shRNA-treated cells were positively correlated with each

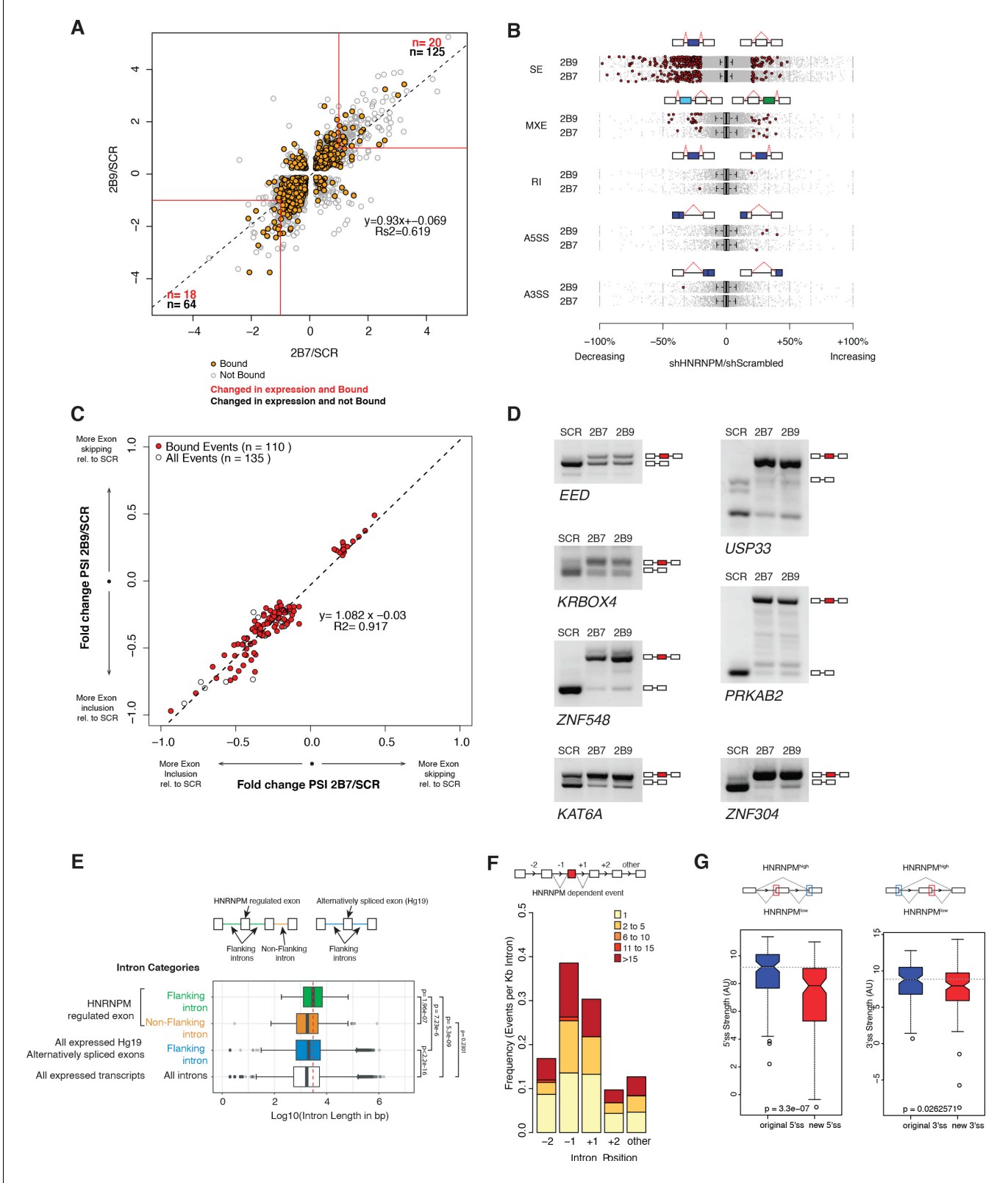

**Figure 4.** HNRNPM depletion results in increased exon inclusion. (**A**) Scatterplot of all significantly changed (p<0.05, FDR < 0.05) genes (gray point) in HNRNPM shRNA-treated cells, and their HNRNPM-binding status. HNRNPM-bound genes are highlighted in orange. Red boxes denote a twofold change cutoff. The total numbers of differentially expressed genes (p<0.05, FDR < 0.05, fold change ≥ 2) are indicated in black, and the numbers of genes that are differentially expressed and bound by HNRNPM are indicated in red. (**B**) Plots showing all alternative splicing events occurring in

*Figure 4 continued on next page*

*Figure 4 continued*

HNRNPM shRNA-treated cells. All unique captured events are plotted in gray, while significantly changing events are shown in red. Change in percent spliced in (ΔPSI; HNRNPM shRNA [2B9] versus scrambled shRNA) for the captured events is shown on the x axis. RI: retained intron; A3SS: alternative 3′ splice site usage; A5SS: alternative 5′ splice site usage; MXE: mutually exclusive exons; SE: skipped exon. (C) Scatterplot of splicing events that are significantly changed (p<0.05, FDR < 0.05, |ΔPSI| > 0.2) in both HNRNPM shRNA (2B7 and 2B9) conditions. ΔPSI is calculated relative to scrambled shRNA-treated cells (SCR). Shown in red are events that occur in transcripts that are bound by HNRNPM. (D) RT-PCR validation of seven representative differentially spliced events that occur in HNRNPM-depleted cells. (E) Distribution of unique intron lengths in the introns flanking linear mis-splicing events (green) compared to that of other introns in the same transcript (orange). Shown also are distribution of lengths in introns flanking other alternatively spliced exons (blues) or all introns within expressed genes. Introns are plotted as Log10(length in base pairs). p-values were determined by Wilcoxon test. (F) Distribution of the numbers of HNRNPM-binding sites across the indicated introns in transcripts with linear mis-splicing events. (G) 5′ (right) and 3′ (left) splice site strength as compared to original, flanking splice sites in HNRNPM-dependent linear transcripts.

The online version of this article includes the following source data for figure 4:

**Source data 1.** Gene expression changes in all HNRNPM-bound genes.
**Source data 2.** Binding status of all significantly differentially expressed genes in HNRNPM knockdown.
**Source data 3.** Binding status of significantly changed skipped exon events.

other (*Figure 4A*), suggesting that the majority of the effects observed on the transcriptome were specific to a reduction in HNRNPM levels. Expression analyses revealed that (*Venter et al., 2001*) only 0.58% (38 of 6510 mapped and bound genes) of HNRNPM-bound RNAs varied significantly (p<0.05, FDR < 0.05, fold change ≥ 2 in both shRNA conditions) in expression levels during HNRNPM depletion in either knockdown condition (*Figure 4—source data 2*), and that (*Wahl et al., 2009*) a minority (19.04%; 38 of 189 differentially and significantly expressed) of transcripts that were significantly altered in expression during HNRNPM knockdown were also bound by HNRNPM (*Figure 4A*, *Figure 4—source data 2*). These data suggested that the majority of transcript abundance changes occurring in these cells were not likely to be a direct consequence of HNRNPM depletion.

## Loss of HNRNPM results in increased exon inclusion

Alternative splicing of transcripts can give rise to events such as changes to alternative 3′ or 5′ splice site usage, cassette exon inclusion or exclusion or intron retention. To determine if some of these events were altered in HNRNPM-depleted cells, we also performed splicing analyses of our RNA sequencing dataset. These analyses revealed that the majority of significant splicing events affected by HNRNPM loss (p<0.05, FDR < 0.05, ΔPSI > 0.2) were the increased inclusion of cassette exons (*Figure 4B*). We observed few significant changes in alternative 5′ or 3′ splice site usage, in intron retention or in mutually exclusive exon usage in either shRNA conditions (*Figure 4B*).

As skipped exon (SE) events were the dominant alteration in HNRNPM depletion, we focused on this group of events for further analyses. We found a core group of 135 SE events that were present and significantly altered in both *HNRNPM* KD/shRNA conditions. 110 (81.5%) of the core 135 identified events were found in genes that were bound by HNRNPM (*Figure 4C*, *Figure 4—source data 3*). Of these 110 events, 96 (87.3%) were events where the cassette exon displayed increased inclusion (lower-left quadrant, *Figure 4C*). We were able to independently validate several of the events (*Figure 4D*) via RT-PCR. Taken together, these data suggested that the majority of splicing changes occurring in HNRNPM-depleted cells were highly likely to be a direct consequence of reduced HNRNPM binding and that HNRNPM depletion tends to result in increased exon inclusion.

To better understand the specificity of HNRNPM towards its target introns, we examined the surrounding genomic and structural features of exons that were differentially included upon HNRNPM depletion. HNRNPM-silenced exons in linear transcripts were flanked by introns that were significantly longer (median: 3024 bp; *Figure 4E*) than other introns in the same transcripts (median: 1826 bp), introns flanking other alternatively spliced exons (median: 2042 bp), or introns in all expressed genes (median: 1728 bp). HNRNPM-binding sites were also preferentially found in these introns compared to other introns of the same transcript (*Figure 4F*). Target exons appeared to bear weaker 5′ and 3′ splice sites compared to original flanking splice sites (*Figure 4G*). Overall, we can conclude that HNRNPM is enriched in the long proximal introns of mis-spliced exons.

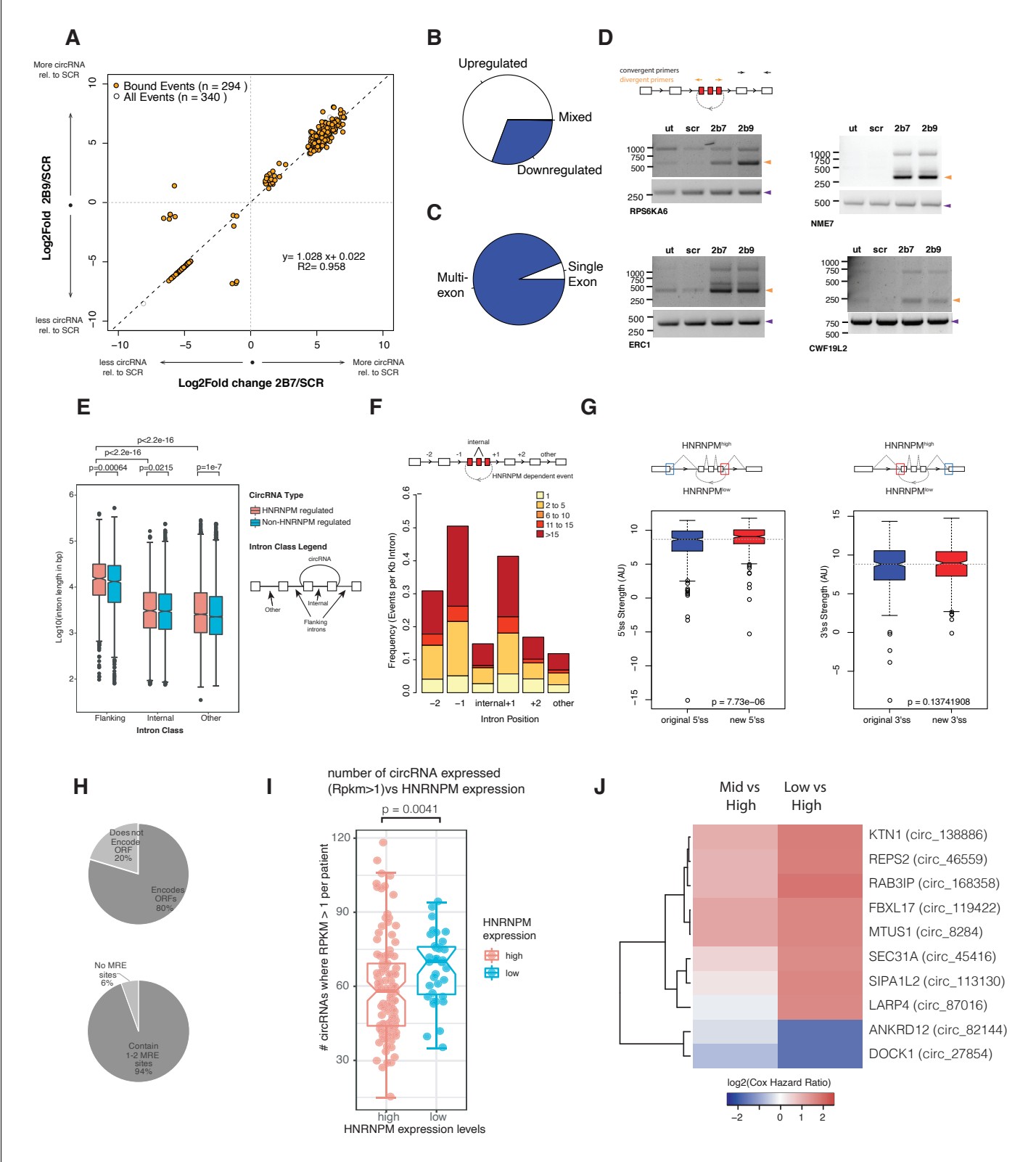

**Figure 5.** HNRNPM depletion results in increased circular RNA (circRNA) formation. (**A**) Scatterplot of circRNA events that are significantly (p<0.05, FDR < 0.05) changed in sh2B7 or sh2B9-treated cells compared to scrambled shRNA-treated cells. All individual captured events are plotted in gray, genes highlighted in orange are bound by HNRNPM. (**B**) Distribution of upregulated and downregulated circRNAs in HNRNPM-deficient cells. (**C**) Distribution of multi-exonic or single-exon circular RNAs regulated by HNRNPM. (**D**) Semi-quantitative RT-PCR validation of four representative circRNA

*Figure 5 continued on next page*

*Figure 5 continued*

events occurring in sh2B7 or sh2B9-treated cells. (**E**) Distribution of unique intron lengths in the introns flanking circular, HNRNPM-dependent and -bound (red) backsplicing events compared to that of other introns (internal or other) in the same transcript. As a reference, intron lengths of circRNAs that are not regulated and not bound by HNRNPM are shown in blue. Introns are plotted as Log10(length in base pairs). p-values were determined by Wilcoxon test. (**F**) Distribution of the numbers of HNRNPM-binding sites across the indicated introns in transcripts with circular mis-splicing events. (**G**) 5′ (right) and 3′ (left) splice site strength as compared to original, flanking splice sites in HNRNPM-dependent circular transcripts. (**H**) Distribution of HNRNPM-dependent circRNAs that encode ORFs or have miRNA-response elements. (**I**) circRNA expression levels in patients that express high or low levels of HNRNPM. (**J**) Heatmap showing the log2(Cox regression hazard ratios) of patients expressing low or mid levels of the indicated circRNAs when compared to patients expressing high levels of the same circRNA. Shown only are the HNRNPM-regulated circRNAs that significantly (p<0.1) predicted disease-free survival in patients with prostate cancer.

The online version of this article includes the following source data and figure supplement(s) for figure 5:

**Source data 1.** Binding status of significantly changed circular RNAs.

**Figure supplement 1.** A subset HNRNPM-regulated circular RNA (circRNA) significantly predicts patient survival (associated with *Figure 5*).

## Loss of HNRNPM results in increased circRNA formation

Aside from regulating alternative splicing events in linear mRNA, the splicing machinery is also known to catalyze the formation of exon-back splicing events in pre-mRNA to generate circRNAs. While the biological functions for most circRNAs remain unclear, their expression is highly regulated and specific to different tissue and cell types (*Memczak et al., 2013*; *Hansen et al., 2013*).

To determine if HNRNPM regulates circRNA biogenesis, we analyzed our RNA-sequencing dataset for backsplicing events. We identified a total of 1357 circularized transcripts in LNCAP cells amongst which 332 circRNAs showed significantly altered expression. More circRNAs (230/332) were upregulated in HNRNPM-deficient cells, suggesting that HNRNPM typically inhibits the formation of these RNAs (*Figure 5A, B*, *Figure 5—source data 1*). The vast majority of these circRNAs (94%; 312/332) were multi-exonic (*Figure 5C*). We were able to validate several of these events in RNaseR-treated RNA with RT-PCR using divergent PCR primers (*Figure 5D*). Similar to our observations for the linear-splicing events, we also observed that the majority (89%; 294/332) of transcripts with changes in backsplicing were bound by HNRNPM (*Figure 5A*), suggesting that the differences in backsplicing frequency observed in HNRNPM-depleted cells are likely to be a direct consequence of reduced HNRNPM binding.

Similar to linear transcripts, introns flanking HNRNPM-silenced circularizing exons (median: 15.3 kb) were significantly longer than other introns (median: 2.5 kb) in the same transcripts or across introns flanking other circRNAs (median: 13.2 kb; *Figure 5E*). HNRNPM-binding sites were also enriched in the proximal 5′ and 3′ introns flanking differentially circularized exons. Fewer HNRNPM-binding sites were found in distal introns or introns within the circularized transcripts (*Figure 5F*). However, HNRNPM-dependent backspliced exons bore stronger 5′ splice sites than that of the original flanking proximal exons (*Figure 5G*).

Finally, we assessed if HNRNPM-dependent circRNAs would contribute to PCa progression. CircRNAs have been reported to regulate gene expression through microRNA (miRNA) response elements as miRNA sponge, through proteins encoded in their sequences. CircRNAs regulated by HNRNPM had few miRNA binding sites, suggesting that they were not likely to perform as miRNA sponges because most of them contained only 1–2 common miRNA response elements (MREs; *Figure 5H*). The majority of HNRNPM-dependent circRNAs (231/294) were however predicted to have coding potential (*Figure 5H*).

To then better understand the physiological relevance of these circRNAs, we took advantage of a recently published dataset (*Chen et al., 2019*) that analyzed circRNA expression in a cohort of 144 PCa patients. 294 of the HNRNPM-regulated circRNAs were found expressed in patients. Expression of these circRNAs was also correlated with *HNRNPM* expression. Indeed, patients that had expressed higher numbers of these circRNAs had significantly lower levels of *HNRNPM* (*Figure 5I*). Interestingly, low expression of a cluster of HNRNPM-regulated circRNAs was significantly correlated with increased risk for patient biochemical relapse (BCR) (*Figure 5J* and *Figure 5—figure supplement 1A*), and have a 'tumor-suppressive' function. This was not likely to be related to gene expression as we did not observe similar effects when patients were stratified by total expression levels of the same genes (*Figure 5—figure supplement 1B*). This was in line with our observations that high

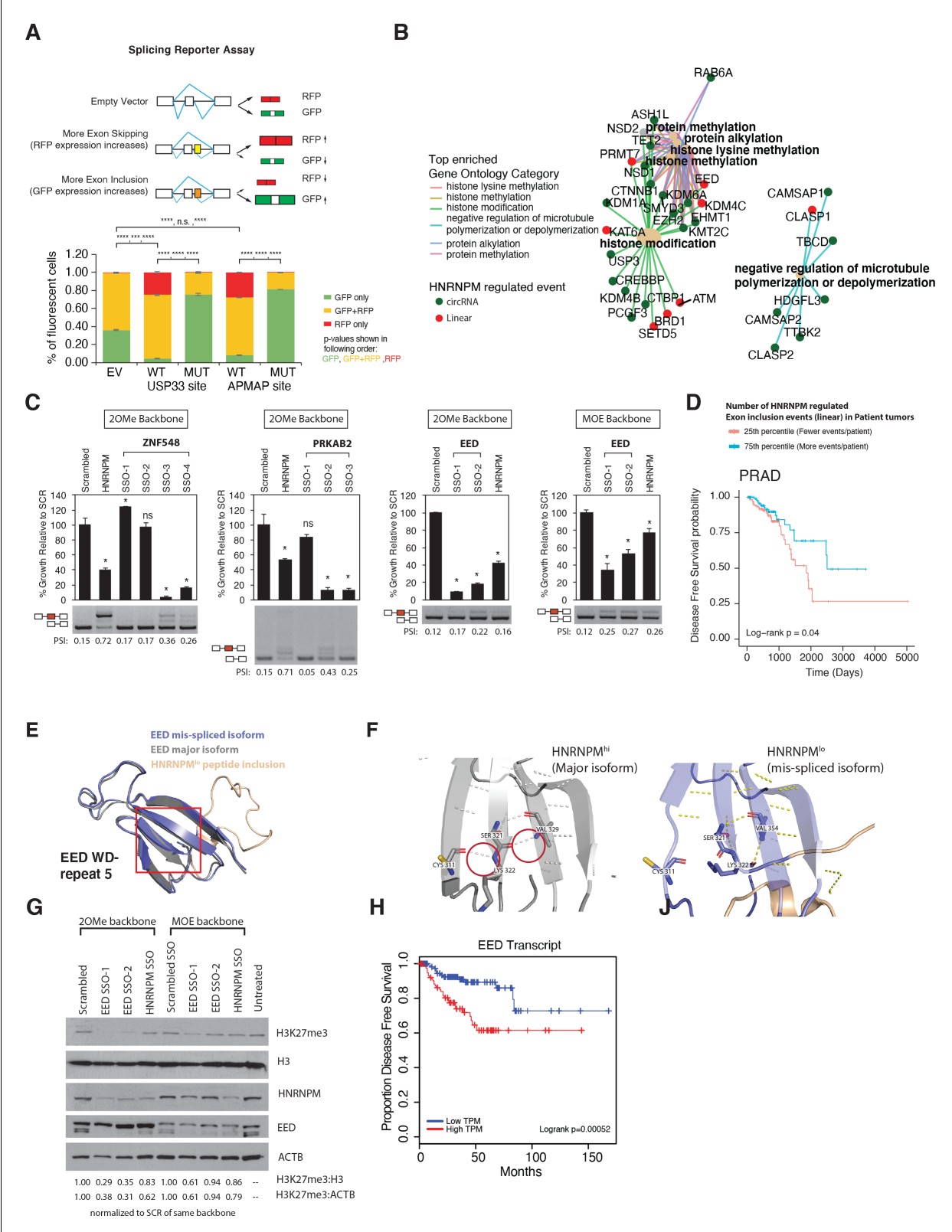

**Figure 6.** Mimicking HNRNPM-dependent linear-splicing events in cells inhibits exon inclusion and cell growth. (**A**, top panel) Schematic showing outcomes of splicing reporter assay. Wildtype (WT) or mutant (MUT) HNRNPM-binding sites at the USP33 or APMAP genes and identified by eCLIP were cloned into a splicing reporter. The percentages of green fluorescent protein (GFP) (indicating exon inclusion) and dsRed fluorescent proteins (RFP) (indicting exon skipping) single- (green and red bars) and double-positive cells (yellow bars) out of all fluorescent cells are shown in a

*Figure 6 continued on next page*

*Figure 6 continued*

barplot in the lower panel. Shown are the mean and standard deviation of three experiments. Student's t-test was used to calculate p-values between the indicated groups. Ns: not significant; ** p<0.01; **** p<0.0001. (B) Gene ontology analysis of transcripts that are either changed in expression or mis-spliced upon HNRNPM knockdown. Both mis-spliced circular (green dots) and linear (red dots) RNAs are shown. Top enriched gene ontology categories (yellow dots) and the genes present in each category are indicated with colored lines. (C) Cell proliferation and exon inclusion outcomes in splice-switching-antisense-oligonucleotides (SSO)-treated cells as compared to scrambled SSO-treated cells. For proliferation assays, growth is shown relative to cells treated with the scrambled non-targeting SSO. The mean and standard deviation of three biological replicates are shown. p-values for growth were determined using Student's t-test. *p<0.05. A representative splicing gel together with the PSI of the HNRNPM-regulated isoforms are shown below each barplot. As controls, a HNRNPM targeting SSO is also included (lane 2). For each gene (ZNF548, PRKAB2, and EED), multiple SSOs targeting different sequences were used. ZNF548 SSO-1, ZNF548 SSO-2, and PRKAB SSO-1 are control SSOs that do not induce inclusion events. For the EED event, both 2-O-methyl (2OMe) and 2-methoxy-ethyl (MOE)-based SSO backbone chemistries were used to control for potential toxicities due to SSO transfection. SSO backbone chemistries used are indicated at the top of each panel. (D) Disease-free survival plot of The Cancer Genome Atlas (TCGA) prostate adenocarcinoma (PRAD) patients, stratified by the total number of events per patient where the PSI of a given HNRNPM-regulated exon exceeds that of the median PSI within the patient population. Patients with more exon inclusion events (top 75th percentile) are shown in blue, whereas patients with less exon inclusion events (lower 25th percentile) are shown in red. (E) WD40 domain 5 of EED when the HNRNPM-regulated exon 10 is included. Structure of the HNRNPM-regulated WD40 domain in WT EED (gray), superimposed on the predicted structure of the HNRNPM-dependent EED isoform (blue). The new peptide generated by the splicing event is depicted in orange. (F) Closeup view of hydrogen bonding interactions that stabilize the WD40 beta sheet in either WT or the new EED isoform. (G) Western blot of H3K27me3 (EED target), HNRNPM, and EED upon treatment with the scrambled, HNRNPM-targeting SSO (HNRNPM) or SSOs that promote the inclusion of EED exon 10 (EED SSO-1 and SSO-2). Backbone chemistry of the SSOs used is indicated. Untreated cells are included as controls. HNRNPM. Shown also are histone 3 and actin B loading controls. (H) Disease-free survival curves of prostate cancer patients when expression in transcripts per million (TPM) of EED is low (blue) or high (red).

The online version of this article includes the following figure supplement(s) for figure 6:

**Figure supplement 1.** HNRNPM-bound sequences inhibit exon inclusion.
**Figure supplement 2.** Structural characterization of HNRNPM-bound introns.
**Figure supplement 3.** HNRNPM regulates a multigenic splicing program to maintain cell proliferation.

*HNRNPM* levels, which are predicted to reduce the expression of these circRNAs, contribute to poorer prognosis in PCa.

## Structure-forming sequences are enriched in the flanking introns of mis-spliced events

To determine whether HNRNPM binding was indeed sufficient to inhibit exon inclusion, we adopted the use of a bi-chromatic splicing reporter that expresses either green fluorescent protein (GFP) or dsRed fluorescent proteins (RFP), depending on the alternative splicing event that occurs (*Figure 6A*). HNRNPM-binding sites of two of the top mis-spliced events in either the USP33 or APMAP transcripts were cloned into this reporter (*Figure 6—figure supplement 1A*, *Supplementary file 2*). As controls, we mutated predicted HNRNPM-binding motifs (*Figure 3E*) found at each site (*Supplementary file 2*). Flow cytometry analysis of 293T cells transfected with either reporter showed that introduction of an HNRNPM-binding site to the reporter resulted in increased dsRed expression as compared to the empty vector control (*Figure 6A, Figure 6—figure supplement 1B*). This suggests that, as predicted, the presence of a HNRNPM-binding site in the reporter was sufficient to cause exon skipping. Conversely, mutation of the HNRNPM-binding sites in the reporter resulted in increased GFP expression (*Figure 6A, Figure 6—figure supplement 1C*, right panels). Taken together, these data add support for our observations that the presence of HNRNPM is indeed required to suppress exon inclusion in cells.

Overall, our observations suggest that HNRNPM preferentially binds to long introns in its target genes (*Figures 4E* and *5E*), and that loss of binding in these introns results in accumulation of mis-splicing events. In long introns, RNA secondary and tertiary structures, stemming from competitive long- and short-range pairings between short inverted repeat (IR) sequences, have been shown to be partially required for appropriate splice site selection (*Rogic et al., 2008*; *Zhang et al., 2014*; *Jeck et al., 2013*). These pairings can be intra- or inter-intron and may function to shorten effective branchpoint distance or mask cryptic splice sites. Such IR pairs tend to be derived from repetitive elements (short-interspersed nuclear elements [SINEs], long-interspersed nuclear elements [LINEs]). We therefore examined if there was differential association of HNRNPM with these elements in mis-spliced transcripts but not in unaffected transcripts. Approximately 43.7% of HNRNPM-bound peaks

across the transcriptome were associated with at least one class of repetitive element (LINE, SINE, repetitive DNA, or long terminal repeats [LTR]) (*Figure 6—figure supplement 2A*; all binding sites; bottom panel). In contrast, 54.1 and 50.5% of HNRNPM-binding sites found within the proximal flanking introns of mis-spliced exons in linear and circular transcripts intersected with such elements (*Figure 6—figure supplement 2A*; top left and middle panels). Consistent with previous observations, L1 LINEs represented the major class of repeats that HNRNPM peaks were associated with (*Figure 6—figure supplement 2B*; *Kelley et al., 2014*).

Recent studies have also suggested that HNRNPs' interactions with RNA tertiary structures such as RNA G quadruplexes (GQ) can impact splicing outcomes (*Huang et al., 2017*). Given that HNRNPM preferentially binds to GU-rich sequences, we also investigated if sequences found within flanking introns of mis-spliced events have the potential to form GQ structures. In linear transcripts, we observed an increased incidence of potential GQ-forming sequences per kb of intron length in the intron upstream of the mis-splicing events (*Figure 6—figure supplement 2C*; left panel) compared to other introns in the same transcript. In contrast, potential GQ structures were enriched in both the upstream (−1) and downstream (+1) introns flanking mis-spliced circRNA events (*Figure 6—figure supplement 2C*; right panel). GQ, when complexed with hemin, has peroxidase activity that can be detected upon addition of substrate by a maximal absorbance at 420 nm (*Li et al., 2013*). Using this assay, we were also able to confirm that several HNRNPM-bound sequences had the capability to form tertiary GQ structures (*Figure 6—figure supplement 2D*). Examination of local secondary structure (±1 kb) by DMS-MaP sequencing around representative HNRNPM-binding peaks in the flanking introns of five mis-splicing events in GMPR2, PRKAB2, USP33, ZNF548, and ZNF304 also revealed the presence of many local structures at and around HNRNPM-binding sites (*Figure 6—figure supplement 2E–G*). Formation of these structures however did not appear to be HNRNPM-dependent (*Figure 6—figure supplement 2G*).

## Mimicking the inclusion of HNRNPM-silenced poison exons by splice switching antisense oligonucleotides inhibits cell growth

Changes in alternative splicing can affect transcript fate and function (*Yang et al., 2016*; *Ellis et al., 2012*; *Naro et al., 2017*; *'t Hoen et al., 2011*; *Lareau and Brenner, 2015*). This, in turn, leads to changes in both the cell's transcriptome and proteome, manifesting as phenotypic transformations. Since changes in alternative splicing, but not gene expression, were the main impact of HNRNPM depletion, we focused on the exon inclusion and circularization events that were changed upon HNRNPM knockdown. Gene ontology analysis of these genes suggested that most of these genes are involved in specific key cellular processes (*Figure 6B*), including histone lysine methylation and the regulation of microtubule polymerization.

The majority of HNRNPM-dependent linear-splicing events were predicted to result in nonsense-mediated decay (NMD) or alternative reading frames in their target protein (*Figure 6—figure supplement 3A*), suggesting that the functions or protein levels of many of these targets were likely to be altered in HNRNPM$^{low}$ cells. To determine if the mis-splicing of some of these genes contributed to the reduction in cell growth, we designed splice switching 2-O-methyl antisense oligonucleotides (SSOs) to induce three HNRNPM-specific exon retention events in HNRNPM-sufficient LNCAP cells. Transfection of SSOs targeting splicing events occurring in *EED*, *PRKAB2*, and *ZNF548* recapitulated splicing events occurring in HNRNPM-deficient cells (*Figure 6C*) and led to the loss of cell viability in target cells. As a control, we used an SSO against HNRNPM that is predicted to result in NMD of the HNRNPM transcript. To rule out the possibility that reduced cell proliferation was due to toxicities of SSO chemistry, we either designed SSOs that targeted our introns of interest, but did not induce switching (ZNF548, SSO-1, and PRKAB2 SSO-1; *Figure 6C*) or *Wahl et al., 2009* used a different backbone chemistry (2'-O-methoxyethyl) that is known to have reduced toxicity and increased binding affinities (*Teplova et al., 1999*; see EED, MOE backbone, *Figure 6C*). In the former case, we saw specific reduction in growth only in cells that exhibited the splice-switching event (*Figure 6C*). In the latter case, we found that growth was inhibited in SSO-treated cells, regardless of the SSO backbone used (*Figure 6C*).

To determine if increased inclusion of HNRNPM-regulated exons is similarly predictive in patient samples, we obtained the PSIs of HNRNPM-dependent exons for patients within the PRAD TCGA dataset. We found that patients that expressed more HNRNPM-regulated exons at PSI higher than the population median had significant survival benefit over those who expressed less (*Figure 6D*).

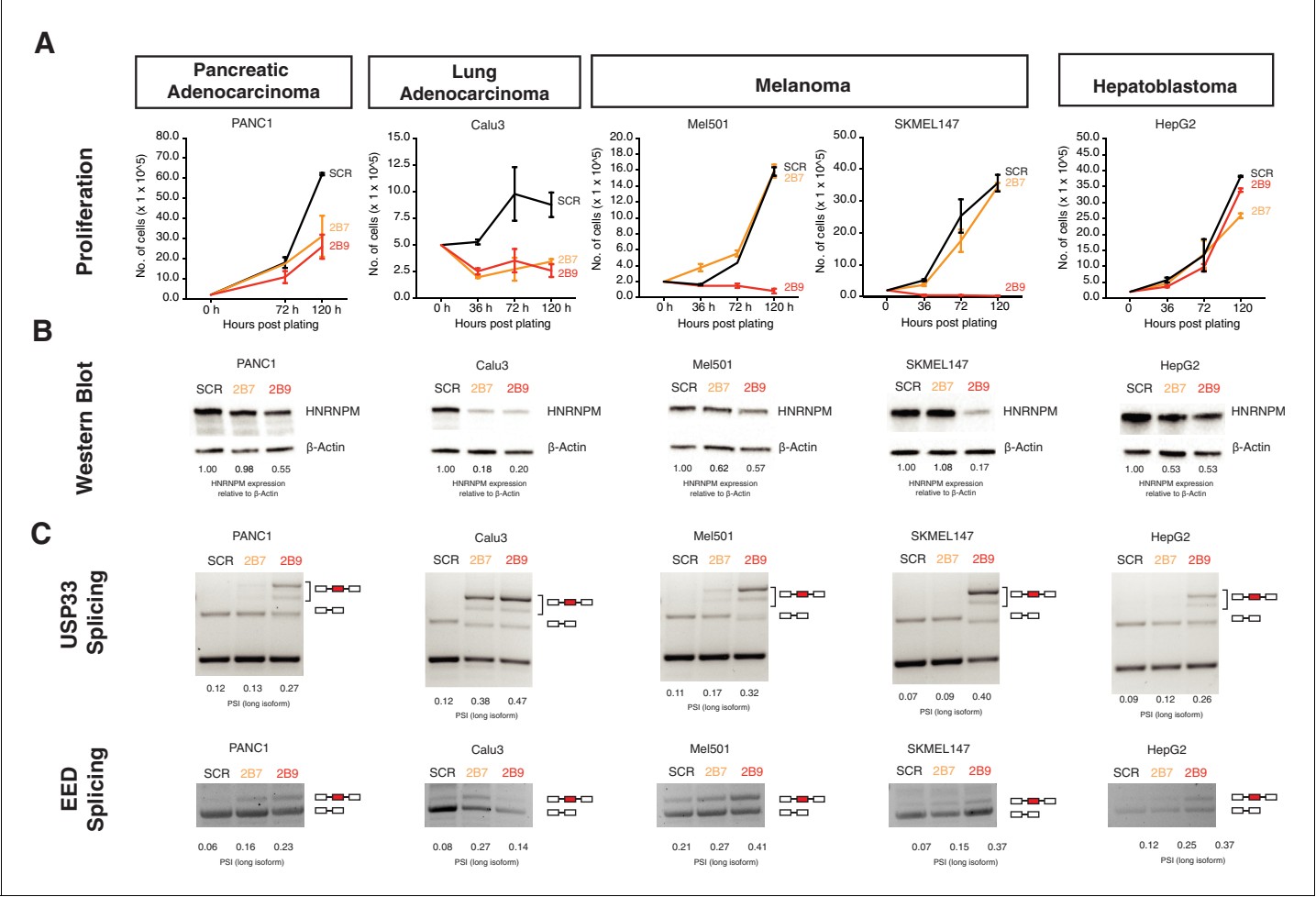

**Figure 7.** HNRNPM regulates growth and splicing in other cancer cell lines. (A) Proliferation curves of PANC1, Calu3, Mel501, SKMEL147, and HepG2 cells, representing cell lines derived from pancreatic adenocarcinoma, lung adenocarcinoma, melanoma, and hepatoblastoma, upon treatment with scrambled (SCR; black) or HNRNPM-specific (2B7; orange or 2B9; red) shRNAs. Shown are the mean and standard deviation of two biological replicates. (B) HNRNPM expression in the indicated cell lines, upon SCR or HNRNPM-specific (2B7 or 2B9) shRNA treatment. ImageJ quantification of HNRNPM protein levels relative to β-actin is shown below. (C) Effect of HNRNPM knockdown on the USP33 and EED transcripts in the indicated cell lines. PSI of HNRNPM-regulated exons, as quantified by ImageJ, is shown below each lane.

The online version of this article includes the following figure supplement(s) for figure 7:

**Figure supplement 1.** HNRNPM mRNA levels are negatively correlated with target circular RNA (circRNA) levels in melanoma short-term cultures.

Overall, these results suggest that the reduction in cell growth in HNRNPM-deficient cells is likely a result of perturbing multiple genes across different pathways, rather than from a single target.

## EED isoform 2 is a novel splicing target of HNRNPM

While HNRNPM loss resulted in NMD events, a significant portion of them also resulted in alternative proteins (*Figure 6—figure supplement 3A*). One such event occurred in the *EED* transcript, which our SSO data suggested was growth inhibitory. We were interested in understanding why this was so. Loss of HNRNPM expression resulted in increased inclusion of *EED* exon 10, resulting in a 25-amino acid added in-frame to the EED protein. To determine how this may affect EED function, we analyzed and modeled the structure of EED protein with or without the peptide extension. Wildtype EED contains seven copies of the WD-repeat motif and folds into a seven-bladed beta-propeller structure. We found that addition of the peptide sequence into the structure results in disruption of hydrogen bonding within the beta sheet of WD repeat 5 (*Figure 6E, F*, *Figure 6—figure supplement 3B*). This is likely to disrupt the stability of the WD repeat and affect EED function. In

support of this, we find that induction of this specific isoform of EED in LNCAP cells is correlated with reduced H3K27me3 activity (*Figure 6G*, *Figure 6—figure supplement 3C*). These results suggest that loss of EED function is likely to contribute to the reduced proliferation we observed upon HNRNPM. Indeed, analyses of the TCGA datasets suggest that patients with reduced EED expression have improved disease-free survival compared to patients with high EED expression (*Figure 6H*).

## HNRNPM regulates growth in other cancers

Finally, we asked if the effects we see with HNRNPM for PCa could be further extended to other cancers with different cell of origins. Assays conducted in Calu-3, Panc-1, SKMEL147, and MEL501 cell lines, representing lung adenocarcinoma, pancreatic adenocarcinoma, and melanoma, respectively, showed that HNRNPM loss resulted in growth inhibition (*Figure 7A, B*), similar to what we had observed with PCa. In addition, loss of HNRNPM in these four cell lines similarly resulted in the same mis-splicing event in *USP33* and *EED* in a dosage-dependent manner (*Figure 7C*), suggesting that HNRNPM similarly regulates splicing events across different cell types. Interestingly, the same assay did not impact viability of HepG2, a hepatoblastoma cell line, despite promoting similar HNRNPM-dependent downstream splicing perturbation (*Figure 7A–C*, rightmost panels). Finally, given the effects that we observed in the melanoma lines SKMEL147 and MEL501, we additionally validated the effects of HNRNPM expression on circRNAs in an independent dataset available to us (*Hanniford et al., 2020*). We confirmed that the expression of the HNRNPM-dependent circRNAs, which we identified in PCa, is significantly negatively correlated with *HNRNPM* expression (R = −0.42, p<2.2e-16) across 10 different primary melanoma cell lines (*Figure 7—figure supplement 1A*).

All in all, our results confirm that HNRNPM may play an important role in regulating growth across multiple, but not all, cancer types.

## Discussion

Recent evidence suggests that cancer cells may be especially vulnerable to disruptions in their splicing machinery compared to normal, untransformed cells (*Koh et al., 2015*; *Hsu et al., 2015*; *Braun et al., 2017*). This is thought to be a consequence of the increased proliferative rates and transcriptional outputs of cancer cells, resulting in 'addiction' to a highly functioning spliceosome. Despite this, splice signatures can vary across different cancers (*Sebestyén et al., 2015*), suggesting that unique or sets of splicing factors may regulate different splicing programs across different cancer types. For instance, ESRP1 has been suggested to drive both oncogenic (*Fagoonee et al., 2017*) and tumor-suppressive (*Ueda et al., 2014*; *Shapiro et al., 2011*) programs in the context of different cancers. Another important point to consider is that splicing factors involved in oncogenesis may also play essential roles in normal cells. Indeed, while we were able to validate known oncogenic splicing factors such as SRSF1 and SRSF2 in our in vivo and in vitro shRNA screens, we have found that short-term depletion of these factors inhibits normal prostate cell growth. Identification of key factors that underpin a cancer-type-specific splicing program, without disrupting normal cell function, is, therefore, important in the development of downstream therapeutics.

Here, we have identified HNRNPM as a potential vulnerability in PCa (*Figure 1*). HNRNPM-depleted LNCAP PCa cells fail to expand both in vitro and in vivo (*Figure 2*). Our data suggest that the arrest in cell proliferation in HNRNPM-depleted cells is likely a consequence of mis-splicing as opposed to an alteration of transcript abundance. Indeed, using splice-switching antisense oligonucleotides to mimic several of these splicing events in cells that expressed HNRNPM was sufficient to inhibit cell proliferation and growth (*Figure 6C*). In support of this, we modeled an exon inclusion event in the EED transcript that suggests that destabilization of one of its WD repeats likely contributes to reduced PRC2 complex activity in PCa cells (*Figure 6E–G*). We have also showed that the majority of circRNAs regulated by HNRNPM are expressed in PCa patients, and that their expression levels are anti-correlated with those of HNRNPM (*Figure 5I*). Reduced expression of some of these circRNAs correlated with poorer disease-free survivals in these patients (*Figure 5J*). Taken together, HNRNPM likely regulates PCa cell proliferation by suppressing exon inclusion and circularization of multiple transcripts across many key homeostatic pathways in cells (*Figure 7—figure supplement 1B*).

Interestingly, the phenotypic manifestation of HNRNPM loss in PCa cells is different from what was previously described. Here we describe a role for HNRNPM in maintaining cell fitness but did not observe changes in cell identity programs in PCa cells. In breast cancer cells, loss of HNRNPM inhibits epithelial-mesenchymal transition (EMT), but not cell growth (*Xu et al., 2014*). In addition, we have shown that cells derived from other cancer types (PANC-1, Calu3, melanoma) are sensitive to HNRNPM knockdown (*Figure 7*). These data are consistent with a context-dependent function for HNRNPM in cancer, which might depend on its downstream targets and cell of origin. Interestingly, the herein identified downstream target EED controls cell proliferation and EMT in context-dependent manner (*Shi et al., 2013*; *Serresi et al., 2016*). Our experiments with induced exon 10 inclusion in EED add a further layer to these observations because they suggest that chromatin templates in the cell can be reprogrammed not only by changing the expression of chromatin factors, but also by altering their splicing. Understanding the extent to which mis-splicing of chromatin factors contributes to establishment of cancer, whether through HNRNPM or other splicing factors, will be important for future studies.

Splicing defects occurring during HNRNPM deficiency primarily involve increased exon inclusion and exon circularization, indicating that HNRNPM normally functions to suppress splice site usage in PCa. This is consistent with previous observations on splicing during HNRNPM loss in other cell types (*Xu et al., 2014*; *Huelga et al., 2012*; *Damianov et al., 2016*). Our data indicate that mis-spliced linear-splicing and circularization events occurring during HNRNPM deficiency are located in exons that are flanked by HNRNPM-bound introns. HNRNPM occupancy is much lower in other introns in the same transcripts, suggesting that its recruitment is important for ensuring appropriate splice site selection.

Members of the hnRNP family of proteins have been shown to be important in ensuring correct splice site selection in target genes. These proteins tend to interact with cis-regulatory splicing silencer sequences in pre-mRNA, and several hnRNPs have also been identified as repressors of splicing (*Matlin et al., 2005*). hnRNP-mediated splicing repression has been attributed to several modes of action, including antagonizing core spliceosomal components and/or splicing enhancer proteins from binding to pre-mRNA (*Zhu et al., 2001*; *Mayeda and Krainer, 1992*; *Mayeda et al., 1993*), as well as alteration of long-range secondary structure in pre-mRNA to inhibit/promote alternate splice site usage (*Solnick and Lee, 1987*; *Blanchette and Chabot, 1999*; *Martinez-Contreras et al., 2006*). These modes of action are not mutually exclusive, and some hnRNPs (e.g., hnRNPA1) have been shown to rely on both, depending on their binding context (*Mayeda and Krainer, 1992*; *Mayeda et al., 1993*; *Blanchette and Chabot, 1999*; *Martinez-Contreras et al., 2006*).

Our analyses suggest that HNRNPM specifically prevents mis-splicing in the context of long genes and is especially enriched in long introns across the genome. Increased intron length has been shown to adversely affect correct splice site selection. For one, cryptic splice sites that compete with the original 5′ and 3′ splice sites are more likely to be found within a longer intron. Secondly, increased intron length extends the delay between RNA polymerase II-dependent synthesis of the 5′ and its associated 3′ splice sites, resulting in a higher probability of RNA secondary and/or tertiary interactions forming to bridge a 5′ss and an incorrect proximal cryptic 3′ss.

Formation of RNA secondary and tertiary structures has been shown to be partially required for correct splice site selection in long introns (*Rogic et al., 2008*; *Zhang et al., 2014*). In vertebrates, large introns bear relatively high densities of repetitive regions that are predicted to form both intra- and inter-intron complementary base pairings (*Shepard et al., 2009*). Such pairings may be long range (>1.4 kb apart; *Aktaş et al., 2017*) and are typically formed between LINEs or SINEs. Competition between such pairs of complementary sequences has been shown to significantly alter splice site selection in both linear and circular transcripts (*Zhang et al., 2014*). It may be that HNRNPM is required to regulate the formation or interactions between these structures at such transcripts in order to maintain correct splice site selection. In line with this, HNRNPM was previously shown to interact with several ATP-dependent RNA helicases (*Brannan et al., 2016*), pointing to a possible association with RNA secondary/tertiary structure. We also observe that HNRNPM is more enriched at repetitive elements in mis-spliced transcripts than other bound transcripts. Long flanking introns that flank mis-splicing events in HNRNPM deficiency also have a higher tendency to form G-quadruplex structures. Interestingly, we do not observe local (<1 kb flanking HNRNPM-bound sites) changes in secondary structure immediately surrounding HNRNPM-bound sites. It could be

that HNRNPM is more important for mediating cross-intron interactions or required for regulating RNA tertiary structures.

Finally, it has been previously reported that long, alternatively spliced genes, bearing elevated intron length to exon length ratios, are highly enriched in pathways associated with cancer and other multi-genic diseases (*Sahakyan and Balasubramanian, 2016*). Understanding how this class of genes is regulated may thus be critical for developing therapeutics against these diseases. Because of their lengths, expression of this class of genes can be rate-limiting for cell survival and/or function. Inhibition of transcription elongation by blocking topoisomerase activity is sufficient to impair their expression (*King et al., 2013*). In a similar way, we have shown that expression of these genes may also be especially vulnerable to depletion of splicing factors such as HNRNPM. One prediction that arises from this observation is that expression of these genes may be modulated through synergistic interactions between inhibitors targeting transcription elongation and splicing factors. Indeed, there is some evidence in the literature pointing towards this. Mutations in the splicing factors SRSF2 and U2AF1 have been shown to impair transcription pause-release and drive R loop formation (*Chen et al., 2018*), suggesting that these two pathways are intimately linked. Identification of other factors similar to HNRNPM that may similarly affect homeostasis of these genes can thus be important for cancer therapy.

## Materials and methods

### Cell culture and treatment

LNCAP (ATCC; RRID:CVCL_0395) and PC3 (ATCC; RRID:CVCL_0035) cells were cultured and maintained in RPMI media supplemented with 10% fetal bovine serum (FBS; Lonza) 1 mM sodium pyruvate (Gibco, 11360070), non-essential amino acids (Gibco; 11140050), and 50 U/ml penicillin-streptomycin (Gibco, 15140163). Primary PrEC (Lonza, CC-2555; RRID:CVCL_0061) were cultured in PrEGM Prostate Epithelial Cell Growth Medium (Lonza, CC-3165) supplemented with BPE, hydrocortisone, hEGF, epinephrine, transferrin, insulin, retinoic acid, triiodothyronine, and GA-1000 as per the manufacturer's recommendation (Lonza, CC-4177). HEK293T (ATCC; RRID:CVCL_0063) cells were maintained in DMEM media supplemented with 10% FBS, 1 mM sodium pyruvate, non-essential amino acids, and 50 U/ml penicillin-streptomycin. SKMEL147 (RRID:CVCL_3876), Mel501 (RRID:CVCL_4633), Calu3 (ATCC; RRID:CVCL_0609), HepG2 (ATCC; RRID:CVCL_0027), and PANC-1 (ATCC; RRID:CVCL_0480) were cultured and maintained in DMEM+ GlutaMAX I (Gibco; 10569-010) media supplemented with 10% FBS (Gibco 10437-028) and 100 U/ml penicillin-streptomycin (Gibco, 15140163). They were cultured on 100 mm × 20 mm Style Cell Culture Treated Nonpyrogenic Polystyrene Dishes (CORNING, 430293). Mel501 and SKMEL147 cell lines were obtained from Dr. Emily Bernstein (MSSM).

Primary PrECs (#24164, #200707 from Dr. David Mulholland, MSSM) were cultured in DMEM+ GlutaMAX I (Gibco; 10569-010) media supplemented with 10% FBS (Gibco 10437-028), Animal-Free Recombinant Human EGF (PeproTech, AF-100-15), A83-01 (TOCRIS, 2939), 100X Insulin-Transferrin-Selenium-Ethanol-amine (Gibco, 51500056), and 100 U/ml penicillin-streptomycin (Gibco, 15140163). They were cultured on CellAdhere Collagen I-Coated 6-well flat-Bottom Plate (STEMCELL Technologies, 100-0362). All cell lines were tested and confirmed to be negative for mycoplasma contamination prior to use.

### Establishment of primary human prostate cultures (#24164, #200707)

Human prostate tissues were obtained from untreated patients with Gleason scores >7 undergoing radical prostatectomy through the Department of Urology (Mount Sinai). To generate patient-derived lines, bulk fresh tissues were digested with collagenase type 1 (Gibco, 1 mg/ml) rotating at 37°C for 2–4 hr. Dissociated cells were plated on tissue culture plates coated with collagen (PureCol, 100 µg/ml) and propagated in human prostate tumor media (DMEM, 10% FBS, 5 mg/ml insulin, 50 ng/ml hEGF, 500 nM A83-01). Plated tumor cells were allowed to grow for multiple passages (>p10) until they were highly enriched with epithelial cells and devoid of stromal cells. The normal tissue counterparts from each patient tumor were cultured in parallel but failed to survive beyond 2–3 passages under the same conditions. All cell lines were tested and confirmed to be negative for mycoplasma contamination prior to use.

## Lentivirus production and infection

For single lentivirus generation, HEK293T cells were seeded to 50% confluency 1 day prior to transfection with the PLKO.1 vector of interest, packaging (d8.9) and envelope plasmids (VSVg) in the ratio of 5:5:1, using Lipofectamine 2000 reagent (Invitrogen). Fresh media was added to the transfected cells after an 18 hr incubation. Subsequently, cell culture supernatant containing lentiviral particles was collected at both 24 hr and 48 hr post media-change, filtered with a 0.22 µm filter, and concentrated by ultracentrifugation at 4°C, 23,000 rpm for 2 hr. For pooled lentivirus generation, PLKO.1 vectors of interest were pooled prior to transfection in HEK293T cells with packaging and envelope plasmids. Ratios of vectors used were maintained as stated in the procedure used for single lentivirus production.

## shRNA pooled screen

The screens were conducted as previously described (*Gargiulo et al., 2014*), with modifications. 5E6 LNCAP cells were seeded 1 day prior to infection in 150 mm dishes and infected with lentivirus pools at a multiplicity of infection of 1 in the presence of 8 µg/ml polybrene. After a 48 hr infection, cells were selected in 1 µg/ml puromycin for 4–5 days. Upon completion of selection, cells were trypsinized and counted. 1.5E6 cells were harvested and saved as the input control for sequencing. The remaining cells were used either for in vitro passaging or in vivo tumorigenesis experiments. For the in vitro passaging experiments, 1.5E6 cells were plated in a 150 mm dish and cells were harvested every 3–4 population doublings, with 1.5E6 cells harvested and plated at each passage. Harvested cell pellets were flash frozen and stored at −80°C. For each injected tumor in the in vivo tumorigenesis experiments, 1.5E6 cells were resuspended in 100 µl of cell culture media mixed with Matrigel in a 1:1 ratio. Cells were then injected into the flanks of 6–8-week-old, male CB17-SCID mice (In Vivos) and allowed to form tumors over time. Mice were monitored for tumor growth every 2–3 days. Tumors were harvested when they attained a size of 400 mm$^3$ and above. Harvested tumors were minced, trypsinized, and ran through a 40 µm mesh to form single-cell suspensions. Cell pellets were then flash frozen and stored at −80°C. Genomic DNA from either the in vitro passaged or the in vivo tumor cell pellets was then harvested with the QIAGEN DNeasy kit in accordance with the manufacturer's protocol. For lentivirus input libraries, RNA was isolated from ~3.5E6 infectious units of lentivirus using a combination of Trizol reagent (Invitrogen) and the Ambion Purelink RNA isolation kit. RNA was converted into cDNA using the Invitrogen Maxima cDNA synthesis reagent. shRNAs were amplified from genomic DNA or cDNA using primers spanning the common flanks of the shRNA expression cassette. These primers included adaptors for Illumina Hiseq sequencing. After PCR amplification, the constituent hairpins in each sample were identified through high-throughput sequencing, and the fold change in hairpin representation was determined by comparing the normalized reads of each sample to the relevant input (P1), plasmid, and viral samples.

## Mice

Mice were housed in compliance with the Institutional Animal Care and Use Committee (IACUC) guidelines. All procedures involving the use of mice were approved by the local Institutional Animal Care and Use Committee (IACUC) and were in agreement with ASTAR ACUC standards.

## Short-term siRNA screen

1E4 PrECs per well were reverse transfected in a 384-well plate with siRNA using Lipofectamine RNAimax reagent (Invitrogen) to a final concentration of 25 nM. siRNAs against PLK1 were used as a positive control, whereas control non-targeting siRNAs were used as negative controls. 96 hr post transfection, total cell yields were measured using the CellTiter 96 AQueous One Solution Cell Proliferation Assay (MTS) as per the manufacturer's instructions. Absorbance readings (490 nm) were taken at 2 hr post incubation with the MTS solution. The relative impact of knockdown on cell growth was then determined by comparing absorbance readings from each sample well to that of the PLK1 or control non-targeting siRNA-treated wells. On-Targetplus human siRNAs SMARTpools were purchased from Dharmacon: HNRNPM: Cat# L-013452-00, HNRNPF: Cat# L-013449-01, HNRNPL: Cat# L-011293-01, SF3A1: Cat# L-016051-00, SCR: Cat# D-001810-10-05; PLK1: Cat# L-003290-00, SRSF1: Cat# L-018672-01, SRSF2: Cat# L-019711-00, and SYNCRIP: Cat# L-016218-00.

## HNRNPM-dependent tumor growth in mice

5E6 LNCAP cells were seeded 1 day prior to infection in 150 mm dishes and infected with lentivirus expressing HNRNPM-specific (2B7, 2B9) or control (Scr) shRNA hairpins at a multiplicity of infection of 1 in the presence of 8 µg/ml polybrene. After a 48 hr infection, cells were selected in 1 µg/ml puromycin for 4–5 days. Upon completion of selection, cells were trypsinized and counted. 1.5E6 cells were harvested and saved as controls for western blot and qPCR. For xenografts, 1.5E6 cells were resuspended in 100 µl of cell culture media mixed with Matrigel in a 1:1 ratio. Cells were then injected into the flanks of 6–8-week-old, male CB17-SCID mice (In Vivos; n = 11 per condition) and allowed to form tumors over time. Mice were monitored for tumor growth every 2–3 days. At the end of the experiment, mice were euthanized and tumors were harvested. Tumors were minced and homogenized in Trizol. RNA was then isolated from tumors, and HNRNPM expression levels verified by qPCR analysis. Primers used for validation were as follows: HPRT1 forward: TGCTGAGGA TTTGGAAAGGG; HPRT1 reverse: ACAGAGGGCTACAATGTGATG; HNRNPM forward: GACCAA TGCACGTCAAGATG; HNRNPM reverse: GTCCTAACCCCATGCCAATAC.

## RNA extraction and RT-PCR validation of linear events

7 of the 135 significantly changed events were used for validation. RNA was isolated using the Purelink RNA Mini Kit (Thermo Fisher Scientific; 12183018A) in conjunction with Trizol Reagent (Invitrogen; 15596018) following the manufacturer's protocol. Following reverse transcription to cDNA with the Maxima First Strand cDNA synthesis kit (Thermo Fisher Scientific; K1641), 30 ng of cDNA was used for each RT-PCR reaction. Primer sequences used for validation are as follows.

### Linear events

> *USP33 forward*: TCCGAGAGCTGAGAAGGAAA;
> *USP33 reverse*: CAATCCTGACAAGTACCAAGGG;
> *ZNF548 forward*: CGAAGTGACGGAACGGAAA;
> *ZNF548 reverse*: GACAGGGCTTTGAAGCTGTA;
> *PRKAB2 forward*: CCATCCTTCCTCCTCATCTACT;
> *PRAKAB2 reverse*: TCCCTTCAAATGGGCTTGTATAG;
> *ZNF304 forward*: GGTTGTGAAGGCTGAGTTCTA;
> *ZNF304 reverse*: GTAGCCACAAGTGCAAAGTTC;
> *EED forward*: GTAGAAGGGCACAGAGATGAAG;
> *EED reverse*: TGAGCAGGAAGACAGTACAAAG;
> *KAT6A forward*: GAGCTCACTGTCTCCAAGATG;
> *KAT6A reverse*: CTGCCCGCTTTATCAGTTTATTC;
> *KRBOX4 forward*: CTAGTTGCGAAGCCAGATGT;
> *KRBOX4 reverse*: TTCAGTGTTTCCTTGCCTATGA.

## RT-PCR validation of circular RNA

4 of the 332 significantly changed events were used for validation. RNA was purified as described above. After RNA isolation, 1 µg of RNA was incubated with 5U of RNaseR (Epicentre) for 45 min at 37℃ in a 14 µl reaction volume to digest all linear RNA. After digestion, RnaseR-treated RNA was reverse transcribed to cDNA using the Maxima cDNA synthesis kit as per the manufacturer's recommendation. Semi-quantitative RT-PCR was then performed on the cDNA using divergent primers to determine levels of circRNA across samples. RT-PCR across constitutively expressed exons was also performed as controls for gene expression levels. CircRNA exon-exon joints were further confirmed by excising PCR-amplified bands in the gel and performing Sanger sequencing. Primers used for validation are as follows.

### Circular events

> *RPS6KA6 circRNA forward*: GAGGAGAGTTACTTGACCGTATTC;
> *RPS6KA6 circRNA reverse*: CCGGGAGAATCTTTAGGTGTTT;
> *NME7 circRNA forward*: GGATCGGGTTAATGTTGAGGAA;
> NME*7 circRNA reverse*: GCTGCATTTCTTATGCCATCTG;
> *ERC1 circRNA forward*: ACGGCTTAAGACACTAGAGATTG;

*ERC1 circRNA reverse*: CGTTCAATTGTCCGCTCTTTC;
*CWF19L2 circRNA forward*: TCACAAGGAGGAAGAAGAAAGAG;
*CWF19L2 circRNA reverse*: TGATAATCTTGGCTCCCAACTT;
*RPS6KA6 linear forward*: AGACAGAGTTCGGACAAAGATG;
*PRS6KA6 linear reverse*: GGGAATGGCCTCTCCTATTTAC;
*NME7 linear forward*: TGTTGCCTGAGTAACCGTATG;
*NME7 linear reverse*: AGGGTCTTGACTGGTGATCTA;
*ERC1 linear forward*: ATGGCCATGGAGAAGGTAAAG;
*ERC1 linear reverse*: CCTGGTCATCGTCCAAGTTATAG;
*CWF19L2 linear forward*: GTGACATGGCTCCCATCTATTT;
*CWF19L2 linear reverse*: CTTCCCTCTGAACCAAGGATTT.

## Western blot analyses

Cells were lysed in RIPA buffer (150 mM sodium chloride, 1% NP-40, 0.5% sodium deoxycholate, 0.1% SDS, 50 mM Tris pH 8.0) for 30 min on ice, before being subjected to sonication (10 s, Setting 4, Microson XL2000). 10–30 µg of protein lysates were ran per lane in 10% SDS-PAGE gels for 1 hr 15 min at 125 V. Proteins were transferred onto 0.22 µm nitrocellulose membranes by wet transfer (Tris-Glycine buffer, 20% methanol) at 30 mA for 1–2 hr on ice. Membranes were blocked in 3% milk, before being incubated in the indicated antibodies overnight. Primary antibody-bound membranes were then washed for 5 min three times in PBS + 0.1% Tween-20, before being probed with HRP conjugated secondary antibodies for 45 min at room temperature. After three washes in PBS + 0.1% Tween-20, membranes were probed with ECL and imaged using film. Antibodies used are as follows: anti-HNRNPM (Santa Cruz, SC-20001; RRID:AB_627740); anti-EED (Millipore 17-10034; RRID:AB_10615775); anti-H3K27me3 (CST 9733S; RRID:AB_2616029); β-actin (CST 5125S; RRID:AB_1903890); tubulin (CST 5346S; RRID:AB_1950376); H3 (Abcam ab1791; RRID:AB_302613). eCLIP experiments were performed as described in *Van Nostrand et al., 2016*, but with the following modifications: two biological replicates were used for these experiments. Approximately 2.5E7 LNCAP cells were grown on 150 mm dishes and UV crosslinked (150 mJ), using the UV Stratalinker 2400. Cell pellets were flash frozen in liquid nitrogen and stored at −80°C for >1 day to facilitate lysis. Lysis was performed in 1 ml ice-cold iCLIP lysis buffer (50 mM Tris-HCL pH 7.5, 100 mM NaCl, 1% Igepal CA630, 0.1% SDS, 0.5% sodium deoxycholate, and 1.1% Murine Rnase Inhibitor [M0214L, NEB]). Cells were allowed to lyse for 15 min on ice, before a short sonication (3 min; 30 s on/30 s off; low setting) in the Diagenode Bioruptor. Sonicated cell lysates were then treated with 40U of RnaseI (Ambion) and 4U of Turbo Dnase (Ambion) for 5 min at 37°C. RNase activity was inhibited with the addition of 1% SuperaseIn reagent, before being clarified with a 15,000 g centrifugation for 15 min at 4°C. Clarified lysates were then subjected to an overnight IP at 4°C using 10 µg of HNRNPM antibody (Santa Cruz sc20001) bound to Protein G dynabeads. 10% input samples were set aside prior to the addition of antibody-bead complexes.

After the overnight IP, immunoprecipitated Protein-RNA complexes were treated with FastAP and PNK treated in the presence of SuperaseIn Rnase Inhibitors. This was followed by the ligation of a 3′RNA linker. The protein-RNA complexes were run on a 1.5 mm 4–12% NuPage Bis-Tris gel at 150 V for 75 min before being transferred on a 0.22 µm nitrocellulose membrane (30 mA, 2 hr). After the transfer, membranes were washed once in PBS. The regions corresponding to approximately 10 kDa below and 75 kDa above HNRNPM-bound complexes were then excised. The region that was excised in this step was further confirmed to contain HNRNPM by running a separate western blot using aliquots of the same lysates. Excised membrane pieces were subjected to proteinase K digestion and RNA isolation in acid phenol/chloroform. The Zymo RNA Clean and Concentrator kit was used to purify the RNA containing aqueous layer from the phenol-chloroform extraction.

After purification, extracted RNA was reverse transcribed, and a 5′ linker was ligated in an overnight DNA ligase reaction. Linker ligated cDNA was then subjected to clean up and amplified using PCR. The final library was purified using 1.8× Ampure XP beads. Adapter dimers were removed using a second Ampure XP bead selection step with 1.4× bead to eluate ratio. Library sizing was checked using the DNA high-sensitivity Bioanalyzer chip, and libraries ranged from 150 to 300 bp in size. Final libraries were pooled and sequenced using the Illumina HiSeq.

## SSO transfections

For proliferation assays, 8E3 LNCAP cells per well were reverse transfected in a 96-well plate with SSOs using the Lipofectamine RNAimax reagent to a final concentration of 100 nM. Control, non-targeting SSOs were used as controls. Total cell yields were measured using the CellTiter 96 AQueous One Solution Cell Proliferation Assay (MTS) at 96 hr post transfection as per the manufacturer's instructions. Sequences of all SSOs used are as follows.

### SCR-SSOs

[2OMe-PS]
mC*mC*mU*mU*mC*mC*mC*mU*mG*mA*mA*mG*mG*mU*mU*mC*mC*mU*mC*mC
[2MOE-PS]
MOErC/*/MOErC/*/MOEdT/*/MOEdT/*/MOErC/*/MOErC/*/MOErC/*/MOEdT/*/MOErG/*/MOErA/*
/MOErA/*/MOErG/*/MOErG/*/MOEdT/*/MOEdT/*/MOErC/*/MOErC/*/MOEdT/*/MOErC/*/MOErC/

### HNRNPM-SSOs

[2OMe-PS]:
mA*mU*mU*mA*mA*mG*mA*mG*mC*mU*mC*mC*mA*mC*mG*mU*mA*mU*mG*mU*mU*mA*mC*mC*mU*mC*mA*mC*mC
[2MOE-PS]:
MOErG/*/MOErG/*/MOErA/*/MOEdT/*/MOErA/*/MOEdT/*/MOErA/*/MOErC/*/MOErG/*/MOErA/*
/MOErA/*/MOEdT/*/MOErG/*/MOErG/*/MOEdT/*/MOErA/*/MOErA/*/MOErA/*/MOEdT/*/MOErA/*
/MOErA/*/MOErG/*/MOErC/*/MOErA/*/MOErC/*/MOErA/*/MOEdT/*/MOErG/*/MOErA/*/MOErA/*
/MOErA/*/MOErA/*/MOErA/*/MOErG/*/MOErA/*/MOEdT/*/MOErG/*/MOEdT/*/MOEdT/*/MOErC/

### EED-SSO-1

[2OMe-PS]
mA*mG*mA*mU*mA*mU*mC*mA*mC*mU*mA*mC*mA*mC*mA*mC*mC*mU*mA*mU*mC*mA*mG*mA*mA*mC*mA*mG*mC*mU*mA*mA*mA*mA*mU*mA*mA*mA*mA*mA
[2MOE-PS]
/MOErG/*/MOErG/*/MOErA/*/MOEdT/*/MOErA/*/MOEdT/*/MOErA/*/MOErC/*/MOErG/*/MOErA/*
/MOErA/*/MOEdT/*/MOErG/*/MOErG/*/MOEdT/*/MOErA/*/MOErA/*/MOErA/*/MOEdT/*/MOErA/*
/MOErA/*/MOErG/*/MOErC/*/MOErA/*/MOErC/*/MOErA/*/MOEdT/*/MOErG/*/MOErA/*/MOErA/*
/MOErA/*/MOErA/*/MOErA/*/MOErG/*/MOErA/*/MOEdT/*/MOErG/*/MOEdT/*/MOEdT/*/MOErC/

### EED-SSO-2

[2OMe-PS]
mA*mA*mG*mU*mC*mU*mC*mA*mG*mC*mC*mC*mC*mC*mU*mG*mU*mU*mC*mU*mC*mU*mC*mA*mA
[2MOE-PS]
/MOErA/*/MOErG/*/MOErA/*/MOEdT/*/MOErA/*/MOEdT/*/MOErC/*/MOErA/*/MOErC/*/MOEdT/*
/MOErA/*/MOErC/*/MOErA/*/MOErC/*/MOErA/*/MOErC/*/MOErC/*/MOEdT/*/MOErA/*/MOEdT/*
/MOErC/*/MOErA/*/MOErG/*/MOErA/*/MOErA/*/MOErC/*/MOErA/*/MOErG/*/MOErC/*/MOEdT/*

/MOErA/*/MOErA/*/MOErA/*/MOErA/*/MOEdT/*/MOErA/*/MOErA/*/MOErA/*/MOErA/*/
MOErA/

### ZNF458 SSO-1

[2OMe-PS]
mA*mA*mG*mU*mC*mU*mC*mA*mG*mC*mC*mC*mC*mC*mU*mG*mU*mU*
mC*mU*mC*mU*mC*mA*mA

### ZNF458 SSO-2

[2OMe-PS]
mC*mC*mU*mG*mU*mU*mC*mU*mC*mU*mC*mA*mA*mC*mA*mC*mA*
mG*mA*mG*mC*mC*mC*mA

### ZNF458 SSO-3

[2OMe-PS]
mG*mU*mU*mA*mG*mU*mA*mA*mC*mC*mA*mA*mA*mA*mC*mA*mG*
mC*mA*mU*mG*mG*mU*mA*mA*mU*mG*mG*mA*mU*mA*mG*mA*mA*mA

### ZNF458 SSO-4

[2OMe-PS]
mA*mA*mC*mA*mG*mA*mG*mA*mG*mU*mG*mG*mC*mC*mC*
mU*mG*mA*mG*mG*mG*mU*mC*mA*mA

### PRKAB2 SSO-1

[2OMe-PS]
mA*mU*mU*mU*mG*mA*mC*mC*mU*mU*mC*mU*mC*mU*mG*
mA*mG*mC*mC*mU*mU*mA*mG*mU*mU*mU*mC*mU*mU*mU - 3'

### PRKAB2 SSO-2

[2OMe-PS]
mA*mA*mA*mC*mC*mU*mA*mC*mU*mU*mG*mG*mA*mA*mA*mG*
mA*mC*mC*mU*mG*mG*mC*mA*mU*mA*mA*mU*mG*mA*mU*mA*mA*mU*mA

### PRKAB2 SSO-3

[2OMe-PS]
mA*mA*mA*mG*mC*mC*mU*mA*mA*mA*mU*mA*mC*mG*mA*mC*
mA*mG*mA*mA*mG*mC*mA*mA*mG*mG*mC*mA*mG*mG*mU*mA*mA*mA*mC

Since the majority of the splicing events we were interested in were deleterious to cells as compared to the non-targeting controls, reduced concentrations of SSOs were used to lower overall transfection efficiency and recover sufficient RNA and protein for downstream analyses. 3E5 LNCAP cells were reverse transfected with SSOs to a final concentration of 25 nM and harvested at 48 hr post transfection. For protein assays, cell pellets were lysed in RIPA buffer containing protease inhibitors before being briefly sonicated. After sonication, lysates were centrifuged (14,000 rpm, 4℃, 10 min) to remove cellular debris, and the resulting supernatant was quantified and used in western blot assays. RNA isolation was performed as described above.

## Bioinformatics analysis (RNA-seq)

RNA sequencing reads were mapped to the hg19 genome using STAR v2.4.2a (*Dobin et al., 2013*). Differential expression analysis was done using the Cufflinks suite v2.2.1 (*Trapnell et al., 2012*).

Experiments were performed using three biological replicates per condition. Significantly differentially expressed transcripts were defined as transcripts that were expressed with greater than twofold change, adjusted p-value<0.05 in both the 2B9 and 2B7 shRNA-treated cells. Alternative splicing analysis was done using rMATS 3.0.9 (*Shen et al., 2014*), both using the Ensembl release 72 human annotation. Downstream splicing predictions for splicing outcomes (NMD, coding, non-coding) was carried out using SPLINTER (*Low, 2017*). Significantly mis-spliced transcripts were defined as transcripts that had |ΔPSI| > 0.1, p<0.05. circRNA expression in HNRNPM knockdown cells was performed using circExplorer 2.0. Differential analysis was done using Cufflinks suite v2.2.1 (*Trapnell et al., 2012*). Significantly differentially expressed circRNAs were defined as circRNAs that were expressed with greater than twofold change, adjusted p-value<0.05 in both the 2B9 and 2B7 shRNA-treated cells.

## Bioinformatics analysis (eCLIP)

e-CLIP-seq analysis was performed according to the protocol outlined in *Van Nostrand et al., 2016* with several modifications outlined here. Briefly, pseudo paired-end reads were first created from single-end reads using bbmap in the BBTools suite (https://jgi.doe.gov/data-and-tools/bbtools/). The first 6 bp of reads were also removed in the adapter-trimming step to improve alignment quality. Peaks that overlapped between the two replicates were identified using BedTools intersect. ChIPpeakAnno package was then used to annotate the genomic regions in which these peaks were found. For motif analysis, we obtained nucleic acid sequences of individual peaks ± 30 bp in the 5′ and 3′ and used HOMER (*Heinz et al., 2010*) analysis software.

For intron and gene length analysis, we defined all expressed genes as genes with >1 FPKM in all three replicates of the scrambled shRNA-treated LNCAP cells. Intron and gene length of all expressed genes were obtained using the BiomaRt tool in R. The overlap of HNRNPM-binding sites and introns or genes of interest was determined using BedTools intersect. To determine the distribution of HNRNPM-binding sites across specific introns, the total number of non-overlapping binding sites in the target introns was normalized to intron length and displayed as sites per kb of intron. For the prediction of G-quadruplex-forming sites, sequences of introns of interest were input into the QGRS tool (*Kikin et al., 2006*) using default settings. The number of G-quadruplex-forming sites per intron of interest was normalized to intron length and displayed as sites per kb of intron.

## HNRNPM-dependent splicing events in PCa patients

The SpliceSeq database (*Ryan et al., 2016*) was used to obtain PSI values for HNRNPM-dependent exons within the PRAD TCGA dataset. For each identified event, PSIs would be classed as 'high inclusion' in a patient if its PSI ≥ (event population median + 0.1), and 'low inclusion' otherwise. The total number of 'high inclusion' and 'low inclusion' events were then summed up per patient. For survival curves, patients were stratified based on the numbers of 'high inclusion' HNRNPM regulated events they expressed. Disease-free survival of the top (75th percentile) and bottom quartile (25th percentile) of patients was plotted. The Cox proportional hazards model was used to calculate the p-value of the difference in survival curves.

## CircRNA expression and HNRNPM expression in PCa patients

Patient clinical attributes and circRNA expression levels were obtained from *Chen et al., 2019*. A circRNA was considered as highly expressed in a patient if it was expressed at levels that were at the 75th percentile of that of the cohort. To calculate HNRNPM expression levels in different groups of patients, the number of highly expressed circRNAs was first counted per patient. Patients were stratified by those that had most numbers of high-expressed circRNAs (>75th percentile of cohort; group = high), least (≤25th percentile of cohort; group = low), or mid-range (25th–75th percentile of cohort; group = mid). HNRNPM levels for each group were then plotted.

## CircRNA expression and BCR survival in PCa patients

Patients' clinical attributes and circRNA expression levels were obtained from *Chen et al., 2019*. For each circRNA, patients were stratified by those that had high-expression circRNAs (>75th percentile of cohort; group = high), low-expression (≤25th percentile of cohort; group = low), or mid-

expression (25th–75th percentile of cohort; group = mid). Biochemical relapse survival curves and hazard ratios were then calculated using the 'Survival' package in R.

## CircRNA expression and HNRNPM expression in melanoma samples

Melanoma short-term cultures (patient samples surgically excised at NYU Langone Health and then adapted to tissue culture conditions) were processed with the Ribo-Zero rRNA Removal Kit (Illumina) for RNA-sequencing. Paired-end reads were processed with the DCC pipeline (version 0.4.7) in order to identify circRNAs. RNA-seq data is deposited in GEO, accession code GSE138711. For further details, see *Hanniford et al., 2020*. Z-scores were calculated for the circRNA backsplice counts and HNRNPM FPKM using the scale() function in R version 4.0.3. Data visualized and statistics calculated using packages ggpubr, ggplot2, ggsci, and dplyr.

## Splicing reporter assays

Bichromatic fluorescent splicing reporter plasmid (RG6) were previously described in *Orengo et al., 2006* and obtained from Addgene (plasmid #80167). Briefly, RG6 was digested with EcoRI and XhoI. Inserts corresponding to HNRNPM-binding regions for APMAP1 (chr20:24945080–24945163; Hg19) and USP33 (chr1:78224956–78225074; Hg19) were cloned in. Sequences are shown in *Supplementary file 2*. Where necessary, HNRNPM-binding sites were mutated using QuikChange XLII Mutagenesis Kit (Stratagene). For flow cytometry assays, 5 µg of each plasmid was transfected into 5E5 HEK293T cells. 1 day post transfection, cells were trypsinized and strained through a 40 µM filter. Flow cytometry experiments were performed on the BD Fortessa. Only singlet and live cells (as determined by DAPI dye exclusion) were analyzed. To avoid biases in data because of differences in transfection efficiency, non-fluorescent cells were gated out using an untransfected control. The relative percentages of GFP and dsRed single- and double-positive cells amongst all fluorescent cells were then determined. Analysis was performed with FCS Express 7.0. p-values were calculated using Student's t-test.

## In vivo DMS probing

LNCaP cells were trypsinized for 1 min at 37°C, trypsin was neutralized by addition of fresh serum-containing medium, and harvested by centrifugation at 4°C for 5 min. After aspiring medium, cells were washed once in PBS and resuspended in DMS Probing Buffer (10 mM HEPES-KOH pH 7.9; 140 mM NaCl; 3 mM KCl). DMS was pre-diluted 1:6 in ethanol, then added to the cell suspension to a final concentration of 100 mM. Cells were incubated at 25°C with moderate shaking for 2 min. Reaction was quenched by placing cells in ice and immediately adding DTT to a final concentration of 0.7 M. Cells were briefly collected by centrifugation for 1 min at 9600 g (4°C). Supernatant was aspirated, and cells were washed once with a solution of 0.7 M DTT in PBS.

## Isolation of nascent RNA

The pellet from a 75%-confluence 100 mm plate was resuspended in 150 µl of Cytoplasm Buffer (10 mM Tris HCl pH 7; 150 mM NaCl) by pipetting using a cut P1000-tip. 50 µl of Cytoplasm Lysis Buffer (0.6% NP-40; 10 mM Tris HCl pH 7; 150 mM NaCl) were slowly added dropwise, after which the sample was incubated on ice for 2 min. The lysate was then layered on top of a 500 µl cushion of Sucrose Buffer (10 mM Tris HCl pH 7; 150 mM NaCl; 25% sucrose). The sample was then centrifuged at 16,000 g for 10 min (4°C). The supernatant (representing the cytosolic fraction) was discarded, and the nuclei were washed in 800 µl of Nuclei Wash Buffer (0.1% Triton X-100; 1 mM EDTA in 1× PBS), then collected by centrifugation at 1150 g for 1 min (4°C). After removing the supernatant, nuclei were resuspended in 200 µl of Glycerol Buffer (20 mM Tris HCl pH 8; 75 mM NaCl; 0.5 mM EDTA; 50% glycerol; 10 mM DTT) and resuspended by pipetting, then transferred to a clean tube. 200 µl of Nuclei Lysis Buffer (1% NP-40; 20 mM HEPES pH 7.5; 300 mM NaCl; 2 M urea; 0.2 mM EDTA; 10 mM DTT) were then added dropwise, while gently vortexing the sample. The sample was then incubated on ice for 20 min and centrifuged at 18,500 g for 10 min (4°C). The supernatant (representing the nucleoplasm) was discarded, and the chromatin fraction was washed once with Nuclei Lysis Buffer, without letting the pellet detach from the bottom of the tube. After discarding the supernatant, 1 ml of ice-cold TRIzol was added directly to the chromatin fraction, and the sample was incubated at 56°C for 5 min to solubilize chromatin. RNA extraction was performed by adding

200 µl of chloroform, vigorously mixing for 15 s, then centrifuging at 12,500 g for 15 min (4°C). After centrifugation, the upper aqueous layer was transferred to a new tube, and mixed with three volumes of ethanol by vigorously vortexing for 15 s. The sample was then loaded on a Zymo RNA Clean and Concentrator-5 column, and purified following the manufacturer's instructions.

## Targeted DMS-MaPseq

DMS-MaPseq was performed with minor changes to the original protocol (*Zubradt et al., 2017*). Oligonucleotides were designed to amplify regions not exceeding a length of 600 nt. Reverse

transcription was carried out in a final volume of 20 µl. 1 µg of nascent RNA was initially mixed with 2 µl of 10 mM dNTPs and 1 µl of 10 µM gene-specific primers, then denatured at 70°C for 5 min, after which the sample was allowed to slowly cool to room temperature by incubation for 5 min on the bench. Reaction was started by addition of 4 µl 5× RT Buffer (250 mM Tris-HCl, pH 8.3; 375 mM KCl; 15 mM MgCl$_2$), 1 µl 0.1 M DTT, 20 U SUPERase• In RNase Inhibitor, and 200 U TGIRT-III Enzyme. The sample was incubated at 50°C for 5 min, followed by 2 hr at 57°C. Reaction was stopped by addition of 1 µl NaOH 5M. Then the sample was incubated at 95°C for 3 min to detach the reverse transcriptase and degrade the RNA. cDNA cleanup was performed using Zymo RNA Clean and Concentrator-5 columns, following the manufacturer's instructions. PCR amplification of target intronic regions was performed using 10% of the purified cDNA for each primers pair and AccuPrime Taq DNA Polymerase (high fidelity), following the manufacturer's instructions. PCR products were resolved on a 2% agarose gel, the bands of the expected size were gel-purified and pooled in equimolar amounts. The sample was then sheared by 20 cycles of sonication using a Diagenode Bioruptor sonicator (30'' on; 30'' off; power: high). Libraries were prepared using the NEBNext Ultra II DNA Library Prep Kit for Illumina, following the manufacturer's instructions.

## DMS-MaPseq data analysis

Data analysis was performed using the RNA Framework v2.5 (http://www.rnaframework.com) (*Incarnato et al., 2016*) All tools were used with default parameters, except for the *rf-map* tool that was called with the parameter *-mp '–very-sensitive-local'*. The percentage of mutations per base was calculated by normalizing the number of mutations on that base by the frequency of that base in the reference (e.g., the number of mutations occurring on A residues, divided by the total number of As present in the analyzed introns).

## RNA G-quadruplex hemin assays

Hemin assays for G-quadruplex formation were conducted as previously described (*Zheng et al., 2015*). Briefly, RNA oligos corresponding to wildtype or mutated sequences of HNRNPM-binding sites found in the introns of EED, USP33, and PRKAB2 transcripts were ordered from IDT DNA. The sequences of these oligos are found in *Supplementary file 3*. RNA oligos were first suspended to 10 µM in folding buffer (20 mM Tris pH 7.6, 100 mM KCl, 1 mM EDTA). They were then heated to 98°C in a water bath and allowed to come to room temperature through passive cooling. Folded RNAs were incubated with 12 µM of hemin (Sigma H-5533, in DMSO) in 1× Buffer 2 (New England Biolabs) at 37°C for 1 hr. Substrate solution (2 mM ABTS, Sigma A9941; 2 mM hydrogen peroxide, Sigma H1009; 25 mM HEPES pH7.4, 0.2 M NaCl, 10 mM KCl, 0.05% Triton X-100, 1% DMSO) was then added to each mixture. After 15 min, absorbance of each sample was measured from 400 nm to 500 nm using a SpectraMax Plus384 Microplate Reader. Samples containing RNA G-quadruplexes display the characteristic absorbance peak at 420 nm.

## Accession numbers

RNA sequencing and eCLIP data have been deposited in the NCBI GEO repository, accession number GSE113786. Data for circRNA in PCa patients was obtained from *Chen et al., 2019*, NCBI GEO repository accession number: GSE113120.

## Acknowledgements

We thank A Jeyasekharan and M Hoppe for sharing protocols and helpful discussions. We are grateful to the staff at the A*STAR Biological Resource Center for support for animal work, the GIS Genome sequencing team for help with RNA sequencing, and the entire E.G laboratory for critical discussion. This work was supported by the Institute of Molecular and Cell Biology, Agency for Science Technology and Research, Singapore. This study made use of data generated by the TCGA Research Network (https://www.cancer.gov/tcga) in its analyses. EG and DW acknowledge support from NMRC/OFIRG/0032/2017 and NRF-CRP17-2017-06. Research reported in this publication was supported in part by National Cancer Institute of the NIH (R01CA249204) and ISMMS seed fund to EG and Melanoma Research Alliance MRA Team Science Award to EH and EG. SZ was supported by the Lee Kuan Yew Endowment Fund Postdoctoral Fellowship. The authors gratefully acknowledge the use of the services and facilities of the Tisch Cancer Institute supported by the NCI Cancer Center Support Grant (P30 CA196521). MS was supported by a NCI training grant (T32CA078207). GG lab was supported by the MDC, the Helmholtz Association, and the ERC. DM was supported by NCI-R01CA197910.

## Additional information

### Funding

| Funder | Grant reference number | Author |
|---|---|---|
| National Cancer Institute | R01CA197910 | David Mulholland |
| National Medical Research Council | NMRC/OFIRG/0032/2017 | Dave Keng Boon Wee Ernesto Guccione |
| National Research Foundation Singapore | NRF-CRP17-2017-06 | Dave Keng Boon Wee Ernesto Guccione |
| National Cancer Institute | R01CA249204 | Ernesto Guccione |
| Icahn School of Medicine at Mount Sinai | | Ernesto Guccione |
| Melanoma Research Alliance | MRA Team Science Award | Eva Hernando Ernesto Guccione |
| Lee Kuan Yew Endowment Fund | | Simin Zheng |
| National Cancer Institute | P30CA196521) | Megan Schwarz |
| National Cancer Institute | T32CA078207 | Megan Schwarz |

The funders had no role in study design, data collection and interpretation, or the decision to submit the work for publication.

### Author contributions

Jessica SY Ho, Conceptualization, Data curation, Investigation, Visualization, Writing - original draft, Writing - review and editing; Federico Di Tullio, Validation, Investigation, Visualization; Megan Schwarz, Florence Gay, Tommaso Tabaglio, Tim Hon Man Chan, Simin Zheng, Validation; Diana Low, JingXian Zhang, Alcida Karz, Data curation, Visualization; Danny Incarnato, Alexander Hall Hickman, Michela Serresi, Gaetano Gargiulo, Investigation; Heike Wollmann, Validation, Methodology; Leilei Chen, Vladimir Roudko, Sujun Chen, Musaddeque Ahmed, Methodology; Omer An, Visualization; Housheng Hansen He, Benjamin D Greenbaum, Salvatore Oliviero, Ivan Marazzi, Supervision; Karen M Mann, Eva Hernando, Resources; David Mulholland, Generation of reagents and scientific inputs; Dave Keng Boon Wee, Resources, Data curation, Supervision, Funding acquisition, Methodology; Ernesto Guccione, Conceptualization, Supervision, Funding acquisition, Project administration, Writing - review and editing

## Author ORCIDs

Salvatore Oliviero (iD) https://orcid.org/0000-0002-3405-765X
Ernesto Guccione (iD) https://orcid.org/0000-0001-7764-5307

## Decision letter and Author response

Decision letter https://doi.org/10.7554/eLife.59654.sa1
Author response https://doi.org/10.7554/eLife.59654.sa2

## Additional files

### Supplementary files

- Supplementary file 1. Table of shRNAs and sequences used for pooled screens.
- Supplementary file 2. Sequences used for bichromatic splicing reporter.
- Supplementary file 3. Sequences of RNA oligos used for RNA G-quadruplex hemin assays.
- Transparent reporting form

### Data availability

All data needed to evaluate the conclusions in this study are present in the paper and/or its Supplementary Materials. eCLIP and RNA-Sequencing data supporting the findings of this study have been deposited into the National Center for Biotechnology Information (NCBI) Gene Expression Omnibus under accession GSE113786.

The following dataset was generated:

| Author(s) | Year | Dataset title | Dataset URL | Database and Identifier |
|---|---|---|---|---|
| Low DHP | 2019 | HNRNPM-regulated splicing dependencies in prostate cancer | https://www.ncbi.nlm.nih.gov/geo/query/acc.cgi?acc=GSE113786 | NCBI Gene Expression Omnibus, GSE113786 |

The following previously published dataset was used:

| Author(s) | Year | Dataset title | Dataset URL | Database and Identifier |
|---|---|---|---|---|
| He HH | 2019 | RNA-Seq with and without RNase treatment in PCa cell lines | https://www.ncbi.nlm.nih.gov/geo/query/acc.cgi?acc=GSE113120 | NCBI Gene Expression Omnibus, GSE113120 |

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
