## [Decision Letter]

**Acceptance summary:**

This is an elegant and thoroughly performed study on the splicing factor hnRNPM, revealing circuits of post-transcriptional regulation relevant for prostate cancer cell growth. In particular, the study focuses on circular RNAs that are upregulated in HNRNPM deficient cells. Using splice-switching antisense oligonucleotides (SSOs), the authors demonstrate that several HNRPNPM-regulated splicing events can inhibit cell growth in HNRNPM expressing cells.

**Decision letter after peer review:**

Thank you for submitting your article "HNRNPM controls circRNA biogenesis and splicing fidelity to sustain prostate cancer cell fitness" for consideration by *eLife*. Your article has been reviewed by 3 peer reviewers, including Juan Valcárcel as the Reviewing Editor and Reviewer #1, and the evaluation has been overseen by James Manley as the Senior Editor.

The reviewers have discussed the reviews with one another and the Reviewing Editor has drafted this decision to help you prepare a revised submission.

Summary:

Ho et al. report the results of a targeted pooled shRNA screen to identify splicing factors important for prostate cancer cell growth in vitro and in mouse xenografts. One of the hits corresponds to the gene encoding hnRNP M, an RNA binding protein overexpressed in prostate cancer cells lines compared to untransformed prostate epithelial cells. They further demonstrate that HNRNPM knockdown (KD) reduces cell proliferation, colony formation, anchorage independent growth in vitro, as well as tumor-xenograft size in vivo. eCLIP, RNA-seq and mutational analysis in minigenes identify GU-rich hnRNP M binding sites in (long) introns flanking exons that become included upon hnRNP M knock down as well as flanking exons undergoing circularization by back-splicing. Modulation of three hnRNP M-regulated alternative splicing events using splice switching antisense oligonucleotides leads to inhibition of cell growth and in the case of the EED gene to reduction of H3K27me3 modifications, which might contribute to the anti-proliferative effects observed upon hnRNP M knock down.

This is an elegant and thoroughly performed study on the splicing factor hnRNPM, revealing circuits of post-transcriptional regulation relevant for prostate cancer cell growth. There are however some issues that the reviewers feel that should be addressed before publication.

Essential revisions:

1. One important question is the extent to which these observations connect with prostate cancer biology. In the absence of results with an animal model, questions could still be asked by examining human tumors:

– First, are there alterations in hnRNP M levels/activity in prostate cancer? The results of Figure 1E are consistent with this but the comparison is limited to three cell lines, which is far from conclusive for assessing the impact of hnRNP M levels on real tumor samples. Analysis of public datasets of tumor samples (including TCGA) could be helpful in this regard.

– Second, are there splicing alterations consistent with changes in hnRNP M activity in tumor samples? The result of Figure 7G shows that the levels of EED transcript correlate with disease progression, but it is unclear what is the connection between EED transcript abundance and the ratio between EED isoforms controlled by hnRNP M (both isoforms are in frame and therefore inclusion or skipping of the alternative exon is not predicted to affect mRNA levels). Splicing-focused bioinformatic tools for the analysis of patient prognosis (e.g. Psichomics) may be helpful in this regard for EED, PRKAB2, ZNF548 and other targets of hnRNP M. The results of Figure 5J are potentially interesting but given the difficulty to predict functional effects of circRNAs, it is difficult to conceptualize these observations.

2. Is hnRNPM actually differentially required in prostate cancer versus normal prostate epithelial cells? While the authors suggest that hnRNPM may not simply be an essential protein in prostate epithelial cells, the data to support this are not clear. For example, what is the effect of hnRNPM depletion in PrEC cells (or, alternatively, does hnRNPM expression alter growth of PrEC, LNCAP, or PC3 cells)? The data presented in the manuscript are valuable regardless of the conclusion of these experiments but it would be helpful to clarify this point.

3. Figure 7C: judging from the changes in growth rates (y axis in panels for individual events), it seems that SSOs induce stronger effects on cell proliferation while inducing substantially more limited changes an alternative splicing in these target genes, how can this be explained? Are these results suggesting a potential toxicity issue with the SSO chemistry? Are these results replicated? Are the effects on cell growth recapitulated by combining the effects of SSOs targeting EED, PRKB2 and ZNF548? Finally, please discuss the fact that in Figure 7F, AON1 leads to EED protein decrease, but AON2 does not, and AON1 also leads to increased HNRNPM expression, suggesting a potential feedback loop. Can restoration of normal EED expression/splicing rescue the effects of hnRNPM loss in prostate cancer cells? As with most splicing factors, the protean number of splicing targets makes it difficult to ascertain if the effects of hnRNPM depletion are related to a few or a multitude of mis-spliced targets. Given the nice work presented on EED splicing here, it may be interesting to investigate if restoring correctly spliced EED could rescue the effects of hnRNPM loss.

---

## [Author Response]

Essential revisions:1. One important question is the extent to which these observations connect with prostate cancer biology. In the absence of results with an animal model, questions could still be asked by examining human tumors:– First, are there alterations in hnRNP M levels/activity in prostate cancer? The results of Figure 1E are consistent with this but the comparison is limited to three cell lines, which is far from conclusive for assessing the impact of hnRNP M levels on real tumor samples. Analysis of public datasets of tumor samples (including TCGA) could be helpful in this regard.

This is indeed an important point. We had tried to address this question in the original Figure S1B (now Figure 1-Supplement 1). We used the TCGA PRAD dataset to stratify patients by their *HnRNPM* expression levels. We found that, in line with our hypothesis, patients that expressed higher levels of *HnRNPM* had significantly worse disease-free survival (p=0.019 by logrank test) compared to patients with lower levels of *HnRNPM*.

This is not the case for the closely related *HnRNPF*.

In addition, we used the TCGA PRAD dataset to look at the overall levels of *HNRNPM* mRNA expression in normal versus tumor samples. Consistent with our observations in cell lines, we saw that tumor samples expressed significantly more *HNRNPM* transcripts when compared to normal controls.

Finally, we performed knockdowns of HNRNPM in primary patient derived prostate cancer cells and showed that HnRNPM was required for the growth of these cells (Figure 2G). These data suggest that the effects that we see in the LNCAP and PC3 cell lines were physiologically relevant and can be extended to real tumor samples.

We nonetheless agree with the reviewer that the connection to prostate cancer biology was limited in scope. We thus extended the analysis to other tumor cell lines, including Panc-1 (pancreatic adenocarcinoma), Calu-3 (lung adenocarcinoma), HepG2 (hepatoblastoma), SKMEL147 and MEL501 (melanoma), to assess how generalizable was the observed mechanism (Figure 7).

As with the effects we saw in prostate cancer, we found that loss of HNRNPM in multiple cancer cell lines, with different cell of origins also lead to reduced proliferation as well as changes in splicing of HNRNPM-targets. This suggests that the effects of HNRNPM extend beyond prostate cancer. Interestingly, in HepG2, despite observing changes in splicing of hnRNPM targets, we did not observe any effect on cell growth. These results point at the fact that hnRNPM is not an essential gene, but acts more as a context-dependent factor in different cancer types.

Finally, given the effects that we observed in the melanoma SKMEL147 and MEL501 cell lines upon HNRNPM knockdown, we additionally validated the effects on HNRNPM-dependent circRNA in an independent dataset available to us. We found a significant negative correlation (R = − 0.42, p < 2.2e−16) between HNRNPM expression and circRNA expression across 10 primary melanoma cell lines in this dataset (Figure 7-Supplement 1), suggesting that HNRNPM likely has similar functions in melanoma as it does in prostate cancer.

– Second, are there splicing alterations consistent with changes in hnRNP M activity in tumor samples? The result of Figure 7G shows that the levels of EED transcript correlate with disease progression, but it is unclear what is the connection between EED transcript abundance and the ratio between EED isoforms controlled by hnRNP M (both isoforms are in frame and therefore inclusion or skipping of the alternative exon is not predicted to affect mRNA levels). Splicing-focused bioinformatic tools for the analysis of patient prognosis (e.g. Psichomics) may be helpful in this regard for EED, PRKAB2, ZNF548 and other targets of hnRNP M. The results of Figure 5J are potentially interesting but given the difficulty to predict functional effects of circRNAs, it is difficult to conceptualize these observations.

Thank you for the comment. To address this question, we looked at EED exon 10 inclusion in the context of high (>90^th^ percentile) or low HNRNPM mRNA (≤ 90^th^ Percentile) expression in the TCGA data set. Since HNRNPM is generally more highly expressed in tumors compared to normal tissue, we chose this cutoff to distinguish between tumors that expressed HNRNPM at levels closer to normal tissues (“low hnrnpm”) or tumors that expressed HNRNPM at levels greater than that of normal tissues (“high hnrnpm”) (see Author response image 1). Overall, we see a trend where patients with the high levels of HNRNPM tend to have low EED exon 10 inclusion (p=0.144, Mann-Whitney test; Author response image 1). In interpreting these results, it is important to note that the levels of *HNRNPM* in tumor samples do not fall to the levels that we see under knockdown conditions (below “normal tissue expression levels”). We therefore will not expect to see as strong a difference in PSI as seen in a knockdown experiment.

**Author response image 1. respfig1:** Analysis of EED exon10 inclusion in TCGA patient samples. (a) To stratify patients between “high hnrnpm” and “low hnrnpm” expression, we used a 90^th^ percentile cutoff within tumor samples. This cutoff was chosen based on *HnRNPM* expression in normal samples. (b) EED exon 10 PSI levels were examined between patients that had “high hnrnpm” and “low hnrnpm” levels as identified in (a). P values were determined using a Mann-Whitney test.

Using data from the SpliceSeq database(2) we have also looked into patient prognosis with respect to some of the linear splicing events that are regulated by HNRNPM. For EED, we see a trend where patients with higher levels of EED exon 10 inclusion have better survival prognosis (based on a 70^th^ percentile cutoff of PSI levels across all patients), although this is not statistically significant based on a logrank significance test. The same is true for other events (example: ZNF548, see Author response image 2).

**Author response image 2. respfig2:** Increased inclusion of single HNRNPM-regulated exons poorly correlates with improved patient survival. Disease free survival plot of TCGA PRAD patients, stratified by the total number of events per patient where the PSI of the indicated HNRNPM-regulated exon exceeds that of the median PSI within the patient population. Patients with more exon inclusion events (top 75^th^ percentile) are shown in red, whereas patients with less exon inclusion events (lower 25^th^ percentile) are shown in blue.

However, when we stratify patients by taking all HNRNPM regulated events into account and ask if there is a difference between patients who present more (more genes with increased PSI compared to the median) or less exon inclusion events (less genes with an increased PSI compared to the median) within the pool of HNRNPM-regulated genes, we found that patients who express more genes with increased PSIs have a significant survival benefit, compared to patients who do not (see Figure 6D). This aligns with our predictions that the lower HNRNPM levels (which promote exon inclusion events) confer survival benefit in patients. Overall, these data, together with the results that we show in Figure 5J for the circular RNAs, suggest that the effects of HNRNPM depletion are likely an accumulation of multiple mis-splicing events, rather than single driver events.

2. Is hnRNPM actually differentially required in prostate cancer versus normal prostate epithelial cells? While the authors suggest that hnRNPM may not simply be an essential protein in prostate epithelial cells, the data to support this are not clear. For example, what is the effect of hnRNPM depletion in PrEC cells (or, alternatively, does hnRNPM expression alter growth of PrEC, LNCAP, or PC3 cells)? The data presented in the manuscript are valuable regardless of the conclusion of these experiments but it would be helpful to clarify this point.

With regards to PrEC cells, we had used siRNA to reduce HNRNPM levels, and showed that reduced HNRNPM levels did not significantly affect cell proliferation in these cells. This data is shown in Figure 1D. We were unable to use the lentiviral system to test the effect of knockdown on growth in these cells due to technical limitations – these cells have a limited number of population doublings before senescing and lentivirus transduction efficiency was low.

On the other hand, reduction of HnRNPM levels in PC3 cells results in inhibition of growth, similar to what we saw with the LNCAP cells. This data was presented in the original Figure S2 (now Figure 2-Supplement 1). We have now made detailed references to this data in the main text.

3. Figure 7C: judging from the changes in growth rates (y axis in panels for individual events), it seems that SSOs induce stronger effects on cell proliferation while inducing substantially more limited changes an alternative splicing in these target genes, how can this be explained? Are these results suggesting a potential toxicity issue with the SSO chemistry? Are these results replicated?

Aside from the scrambled control SSOs, our experiments with PRKAB2 and ZNF548 also include SSOs (see ZNF548 SSO-1 and PRKAB2 SSO-1 in Figure 6C) that do not induce exon inclusion events as controls. Cells that are treated with these SSOs proliferate to the same extent as cells treated with the scrambled non-targeting ones, suggesting that SSO chemistry used is not toxic to the cells. The results shown in Figure 7C represent 2 independently performed experiments for cell proliferation.

In addition, for EED, we have repeated experiments using a different SSO chemistry. SSOs were synthesized with the MOE backbone compared to the 2-OMe backbone we used. We still observe the same reduction in cell proliferation, as well as changes in splicing, suggesting that the effects of reduced cell growth that we see are most likely sequence specific, rather than chemistry specific (Figure 6C).

Are the effects on cell growth recapitulated by combining the effects of SSOs targeting EED, PRKB2 and ZNF548?

Yes, we have performed the experiment in question, and we see that effects of cell growth is recapitulated by combining the effects of SSOs targeting EED, PRKAB2 and ZNF548 (see Author response image 3). Note that to avoid potential issues of toxicity with due to increased lipofectamine and overall SSO concentrations in this combinatorial experiment, we used a 4-fold lowered concentration of individual SSOs (25nM/SSO/sample) for this experiment as compared to what we have used (100nM/SSO/sample) in experiments we had previously shown. Where necessary, scrambled (SCR), non-targeting SSOs were added to ensure that the total final concentration of SSOs cells were exposed to remained the same.

Overall, due to the lower concentration used, we did not observe strong synergy among the different SSOs. It is likely though that combination of multiple events (there are 135 mis-spliced events, of which 110 are bound by HNRNPM) are indeed causing the phenotype, which is somewhat hard to reproduce in vitro.

**Author response image 3. respfig3:** Combined knockdown of multiple splicing events results in inhibition of cell proliferation. LNCAP cells were transfected with 2OMe SSOs targeting ZNF548 (ZNF548 SSO-3), PRKAB2 (PRKAB SSO-2) and EED (EED SSO-1). To adjust for overall SSO concentrations in cells, control (SCR) targeting SSOs were added to each well as indicated in the table below the barplots. Cell proliferation was measured at 96 hours post transfection with the MTS assay (Promega). Shown are the mean and standard deviation of 3 biological replicates.

Finally, please discuss the fact that in Figure 7F, AON1 leads to EED protein decrease, but AON2 does not, and AON1 also leads to increased HNRNPM expression, suggesting a potential feedback loop.

While the effects of reduced H3K27me3 appear to be consistent with the use of the SSOs targeting EED event and HNRNPM, we have not always observed a direct correlation between EED and HNRNPM protein levels in other replicates, suggesting that a potential feedback loop is unlikely. A second independent replicate of the protein blot, performed alongside samples that were treated with the MOE chemistry is shown in Figure 6G. H3K27me3 levels, normalized to H3 and ACTB is quantified in Figure 6G. The lower EED isoforms are clearly visible, and are depleted by both EED and HNRNPM SSOs.

Can restoration of normal EED expression/splicing rescue the effects of hnRNPM loss in prostate cancer cells? As with most splicing factors, the protean number of splicing targets makes it difficult to ascertain if the effects of hnRNPM depletion are related to a few or a multitude of mis-spliced targets. Given the nice work presented on EED splicing here, it may be interesting to investigate if restoring correctly spliced EED could rescue the effects of hnRNPM loss.

Thank you for the suggestion. As the reviewer suggested, we tried to perform rescue experiments by overexpressing the wild type EED protein in HNRNPM depleted cells. Unfortunately, we were unable to achieve rescue in proliferation, suggesting that the effects of HNRNPM depletion is not solely dependent on misplicing of *EED* alone.

**Author response image 4. respfig4:** EED overexpression in HNRNPM knockdown cells. LNCAP cells were transduced with either a control (empty vector) or EED overexpressing lentivirus (Blasticidin resistant). After selection, cells were transduced with lentivirus expressing either the scrambled (SCR) or HNRNPM-specific (HNRNPM kd; 2B9) shRNAs (Puromycin resistant). After the second selection step, 2E5 cells were plated and proliferation over time was monitored through counting every 24 hours. The mean and standard deviation of 2 independent replicates is shown.